# IFN alpha inducible protein 27 (IFI27) acts as a positive regulator of PACT-dependent PKR activation after RNA virus infections

Darío López-García, Vanessa Rivero, Laura Villamayor, Marta L. DeDiego ![ORCID]*

Department of Molecular and Cell Biology, Centro Nacional de Biotecnología-Consejo Superior de Investigaciones Científicas (CNB-CSIC), Madrid, Spain

* marta.lopez@cnb.csic.es

## Abstract

Protein kinase R (PKR) expression is induced by interferons. This protein is activated by double-stranded (ds) RNAs or RNAs containing duplex regions, produced after different stimuli, such as after viral infections, leading to the phosphorylation of the eukaryotic translation initiation factor 2α (eIF2α), and subsequently inhibiting cellular and viral protein translation. This function may lead to different effects such as to impairing the replication of RNA viruses by inhibiting viral protein translation, and to modulating the innate immune responses after viral infections by affecting the translation of effector proteins. In this work, we identify, for the first time, an interaction of IFN alpha inducible protein 27 (IFI27) with PKR-activating protein (PACT or PRKRA) and with PKR, showing that the interaction of IFI27 with PACT is likely mediated by dsRNAs or RNAs containing duplex regions, and that the interaction of IFI27 with PKR is PACT-dependent. Interestingly, using IFI27 knocked-down, knocked-out and overexpressing tumour-derived, established cells, we show that these interactions trigger a potentiation of the activity of PKR and, therefore, a decrease in protein translation. Moreover, we find that IFI27 increases PKR function in cells infected with different RNA viruses such as Severe Acute Respiratory virus 2 (SARS-CoV-2), and Vesicular Stomatitis virus (VSV), and in cells transfected with the dsRNA analog poly(I:C), suggesting a broad effect of IFI27 on PKR activation. Moreover, we show that IFI27 expression increases the formation of stress granules (SGs) at the cell cytoplasm, correlating with the increased PKR activation mediated by IFI27, as it has been shown that the translational arrest induced by activated PKR leads to the formation of SGs. Mechanistically, we describe that this ability of IFI27 to activate PKR is dependent on its interaction with PACT. Further understanding of the regulation of PKR activity will allow us to develop new antiviral drugs to modulate this signalling axis, which is crucial in RNA virus infections.

**Data availability statement:** The authors confirm that all data underlying the findings are fully available without restriction. All relevant data are within the paper and its Supporting Information files.

**Funding:** This work was supported by grant PID-2021-123810OB-I00, funded by MCIN/ AEI /10.13039/501100011033/ and by FEDER; and by grant CNS2022-135276, funded by MCIN/ AEI/10.13039/501100011033 and the European Union NextGenerationEU/PRTR, to MLD. The project that gave rise to these results received the support of a fellowship from "la Caixa" Foundation (ID 100010434). The fellowship code is LCF/BQ/DR22/11950020 (granted to DLG, who received funds for his salary during three-years from "la Caixa" Foundation)". The funders had no role in study design, data collection and analysis, decision to publish, or preparation of the manuscript.

**Competing interests:** The authors have declared that no competing interests exist.

## Author summary

Protein kinase R (PKR) is a protein whose expression is induced after innate immune responses are triggered. Its activation involves the binding to double-stranded (ds)RNAs or RNAs containing duplex regions, produced after different stimuli, such as viral infections. This PKR activation leads to eIF2α phosphory-lation, and subsequent inhibition of protein translation, that may lead to impair-ing viral replication, and to modulating host innate immune responses. In this manuscript, we identify a completely novel interaction of IFI27 with PACT and with PKR. The interaction of IFI27 with PACT is likely mediated by RNAs, and the interaction of IFI27 with PKR depends on PACT. Importantly, these inter-actions increase PKR activity and, therefore, decrease protein translation after the infection with different viruses. IFI27 increases PKR function dependent on its interaction with PACT, in cells infected with Severe Acute Respiratory virus 2 (SARS-CoV-2), and Vesicular Stomatitis virus (VSV), and in cells transfected with poly(I:C), suggesting that the effect of IFI27 on PKR activity is broad. Fur-ther understanding of the regulation of PKR activity will allow us to develop new molecules to modulate PKR activity.

## Introduction

The immunity against viral infections is composed of several layers, of which the innate immune responses act like the first line of defence. These responses are initi-ated when specialized receptors, the Pattern Recognition Receptors (PRRs) recog-nize several molecules associated to viral infections, known as Pathogen Associated Molecular Patterns (PAMPs). PAMPs include RNA motifs like double-stranded RNA (dsRNA) or 5´-phosphate single stranded RNAs (ssRNAs), glycoproteins or proteo-glycans [1,2]. The recognition of viral RNA motifs is one of the most studied pathways of the antiviral responses, as several cellular proteins are known to recognize these structures and initiate antiviral pathways [3]. One can differentiate viral RNA-specific PRRs such as Toll-like receptors (TLRs) [3,4] and RIG-I like receptors (RLRs) [3,5,6], and viral RNA-dependent effectors, such as RNAseL-olygoadenylate synthetase (OAS) pathway [7,8], adenosine deaminase acting on RNA (ADAR) [9,10] and the protein kinase RNA-activated (PKR) [11,12].

Human PKR is a 551 amino acid-long, serine-threonine kinase, encoded by the Eukaryotic Translation Initiation Factor 2 Alpha Kinase 2 (*EIF2AK2*) gene, located on the chromosome 2 [13]. This gene is constitutively and ubiquitously expressed on mammalian cells, however, this gene is an interferon (IFN)-stimulated gene (ISG), as it has a IFN-stimulated response element (ISRE) in its promoter, and its transcription is induced by IFN [14]. This kinase is a central player of the innate immunity against RNA viruses, and it has been implicated in other cellular processes, such as the reg-ulation of mRNA translation, apoptosis and proliferation. PKR dysregulation has been linked to neurodegeneration, cancer, inflammation and metabolic disorders [13,15].

PKR has two N-terminal tandem dsRNA binding motifs (dsRBMs), and a C-terminal catalytic kinase domain that, on absence of dsRNAs, is autorepressed on a monomeric state [11,16]. When viral dsRNA is recognized and bound by PKR dsRBMs, the catalytic kinase domain is released and activated. Different PKR molecules oligomerize after dsRNA binding, enabling *trans*-phosphorylation on Ser and Thr of different PKR molecules [11,16,17]. Activated PKR then phosphorylates the Ser52 of the eukaryotic translation initiation factor 2α (eIF2α). Phosphorylated eIF2α can no longer initiate new translation cycles, and this leads to a blocked translation initiation [11,18,19]. As translation is blocked, viruses are not able to sustain their protein synthesis, making this antiviral mechanism one of the major host defence factors [20]. However, different viruses have evolved to escape this antiviral pathway, by employing different strategies: sequestration of dsRNA, different translation systems (internal ribosome entry site, IRES), sequestration, degradation or inhibition of the phosphorylation of PKR, or dephosphorylation of eIF2α [19–21].

PKR function can be regulated by viral and cellular RNAs (both activators and inhibitors) as well as by viral proteins. For example, Influenza Virus non-structural protein 1 (NS1), Severe Acute Respiratory Syndrome Coronavirus 2 (SARS-CoV-2) nucleoprotein (N) and non-structural protein 15 (nsp15), Poxvirus E3L and K3L proteins, baculovirus PK2 protein, Hepatitis C Virus NS5A protein or Human immunodeficiency virus I (HIV-I) Tat protein, inhibit PKR functions [22–25]. Furthermore, cellular protein partners such as PKR activating protein (PACT or PRKRA) [26], transactivation response element RNA-binding protein (TRBP) [27,28], tRNA-dihydrouridine synthase 2 (hDUS2) [29], nuclear factor 90 (NF90) [30] or ADAR1 [22,31] modulate PKR functions.

Of the different protein regulators of PKR, one of the best studied is PACT, a protein composed of three dsRBMs, two of which interact with RNA, while the third is involved in protein-protein interactions [22,32,33]. Upon interaction with PKR, the first two PACT dsRBMs interact with PKR dsRBMs, while the third one associates with the kinase domain [22]. PACT, in the presence of stress, is phosphorylated at Ser287 residue, and this was identified as a hallmark of downstream activation of PKR. This phosphorylation of PACT reduces its affinity for TRBP (which acts as an inhibitor of PKR) and increases its affinity for PKR, thus activating the last [32]. PACT has been shown to activate PKR after different types of stress, such as endoplasmic reticulum perturbations or disturbances of the oxidative phosphorylation function of mitochondria, even on absence of viral infections [34]. Thus, the PACT-PKR interaction is well known, but there are still a lot of questions on the implication of other regulators on this axis, especially in the context of different viral infections. For instance, Peters *et al*. [35] described that Us11 protein of Herpes Simplex Virus type 1 is able to bind to PKR, avoiding its activation by PACT. The Orf virus (ORFV, from the *Poxviridae* family)-encoded protein OV20.0 can interact with PACT and block its ability to activate PKR [36]. Furthermore, Clerzius *et al*. [37] showed that during Human Immunodeficiency Virus 1 (HIV-1) infection, PACT becomes a PKR inhibitor instead of an activator, by a mechanism that may involve ADAR1-PACT interaction, but that has not been fully elucidated [37,38]. It is unknown whether PACT is phosphorylated when it acts as an inhibitor of PKR, as changes or absence of its phosphorylation, or even other unknown post-translational modifications, could explain this switch from an activator to an inhibitor of PKR [15].

Therefore, the search for new proteins that may be regulating PKR function is essential, due to the importance of this kinase in inhibiting the replication of RNA viruses of high importance at a socio-sanitary level. Furthermore, PKR could modulate the innate immune responses after viral infections by modulating the translation of effector proteins. In fact, it has been shown that Hepatitis C Virus blocks interferon effector function by inducing PKR phosphorylation, leading to decreased translation of ISGs [39,40] and that translation efficiency of type I IFN and ISGs were significantly downregulated by PKR activation after Zika virus infection [41]. In this context, we show for the first time an interaction of IFN alpha inducible protein 27 (IFI27) with both PACT and PKR. IFI27, also known as ISG12A, is a 122 aminoacid-long, 12 kDa protein that belongs to the FAM14 family [42]. IFI27 has been scarcely studied, however, we have recently shown that this protein is able to negatively modulate the innate immune responses both in vitro and in mice, by interacting with the RLR Retinoic Inducible Gene–I (RIG-I) [43] and melanoma differentiation-associated gene 5 (MDA-5) [44] and negatively

modulating their functions. We have shown that these interactions are likely mediated by its ability to interact with polyinosinic-polycytidylic acid (poly(I:C)), a synthetic analog of double stranded RNAs [43,44].

In this manuscript, we aim to define IFI27 functions, starting from a proteomics analysis of IFI27-binding proteins. With this objective, we identify, for the first time, an interaction of IFI27 with PACT and PKR proteins, which involves a potentiation of the activity of PKR and therefore a decrease in cellular translation. Particularly, we show that IFI27 potentiates PKR function in cells infected with different viruses such as SARS-CoV-2, and Vesicular Stomatitis virus (VSV), and in cells transfected with poly(I:C), suggesting that the effect of IFI27 on PKR activation is broad. We also show that this ability of IFI27 to activate PKR is dependent on its interaction with PACT, and that the interaction of IFI27 with PKR is PACT-dependent. Further understanding of the regulation of PKR activity will allow us to develop new antiviral drugs to modulate this signalling axis, which is crucial in RNA virus infections.

## Results

### IFI27 interacts with PKR and PACT

IFI27 is a small hydrophobic protein, located at the mitochondrial membrane [45], and its expression is highly induced by type I interferons [42,43,46]. We previously published that IFI27 is able to interact with dsRNA [43,44], an ability that is characteristic of several proteins involved in the recognition of viral RNAs and the regulation of the antiviral responses [47]. However, the function of this protein has been scarcely studied. In order to identify possible functions of IFI27 within the innate immune responses against viral infections, IFI27 protein was overexpressed by transfecting a pCAGGS plasmid encoding for IFI27 fused to an HA tag (pCAGGS-IFI27-HA) in HEK-293T cells, or the cells were transfected with a pCAGGS plasmid encoding IFI44 fused to an HA tag (pCAGGS-IFI44-HA), or with an empty plasmid, as control, and these cells were subsequently transfected with poly(I:C), an analog of viral dsRNAs, in order to activate cellular pathways involved in the response to RNA viruses. These cells were lysed, and the protein extracts were incubated with anti-HA antibodies bound to agarose beads to retain IFI27-HA, IFI44-HA, and IFI27 and IFI44-associated proteins in this matrix, and the bound proteins were later identified by mass spectrometry (MS) (Fig 1A). The complete list for the IFI27-bound proteins showing a protein score higher than 40, eliminating the proteins that were also identified in the cells transfected with the empty plasmid (which were considered as non-specific), is shown in the S1 Table. All these proteins with a score higher than 40 were analysed by Gene Ontology, using NIH DAVID software (https://davidbioinformatics.nih.gov/). Proteins were classified by biological functions (S2 Table) and molecular functions (S3 Table), and we selected all those groups of proteins whose function showed a False Discovery Rate (FDR) lower than 0.05. RNA binding (GO:0003723~RNA binding) was the main molecular function identified within the IFI27 interactome, representing 50.6% of all the proteins found, and showing an FDR of 2.27E-22 (S3 Table). This was remarkable, as we have recently described that IFI27 is a novel dsRNA binding protein [43]. RNA binding is closely related to virus detection and translational processes within cells [48–50]), so it was very interesting to find that one of the most overrepresented biological functions within IFI27 interactome was translation (GO:0006412, accounting for 10.8% of found proteins, and with a FDR of 4.1E-4) (S2 Table). Among all the proteins that were part of both terms (Translation and RNA binding, S2 and S3 Tables, respectively), we were particularly interested in analysing the presence of two of them: PKR and PACT endogenous proteins, as these two proteins showed very high scores of 565, and 494, respectively (Fig 1A and S1 Table). As control, PKR and PACT were not detected after immunoprecipitating IFI44-HA in the cells overexpressing IFI44-HA, strongly suggesting that the binding of PACT to IFI27 is not mediated by the HA tag. Exponentially modified Protein Abundance Index (emPAI), which estimates the absolute protein amount by analyzing the number of observed peptides divided by the number of observable peptides per protein [51], was higher for PACT (emPAI 1.92), than for PKR (emPAI 0.71) (Fig 1A), suggesting that the abundance of PACT in the sample was higher than PKR abundance. Interestingly, the protein phosphatase 1C (PPP1CC), encoding the gamma isoform of the phosphatase catalytic subunit, was identified as well, although to lower protein score (279) (S1 Table). PPP1CC dephosphorylates eIF2α [52]. Also, TARBP2 Subunit of

**A**

| protein_accesion | protein_description | protein_score | protein_mass | protein_coverage | emPAI |
|---|---|---|---|---|---|
| A0A087WZR1 | Interferon alpha-inducible protein 27, IFI27 | 400 | 11233 | 67.2 | 17.15 |
| O75569 | Interferon-inducible double-stranded RNA-dependent protein kinase activator A, PRKRA, PACT | 494 | 34750 | 44.4 | 1.92 |
| P19525 | Interferon-induced, double-stranded RNA-activated protein kinase, EIF2AK2 | 565 | 62424 | 26 | 0.71 |

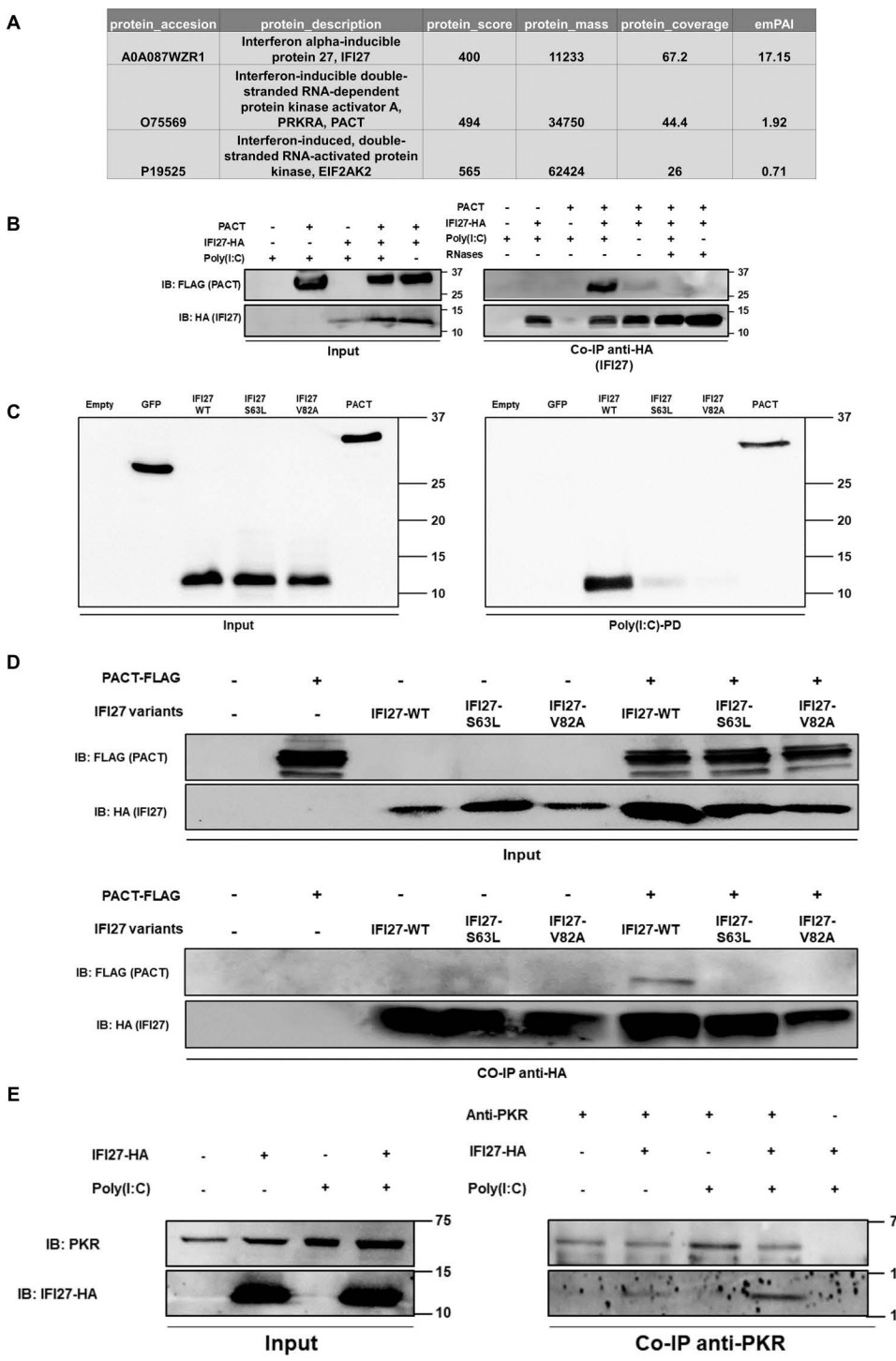

**Fig 1. IFI27 interacts with PKR and PACT. (A)** HEK-293T cells were transiently transfected with a pCAGGS-IFI27-HA plasmid, and 24h later, the cells were transfected with poly(I:C) at 3000 ng/ml. 24h after poly(I:C) transfection, protein extracts were obtained by lysis and were incubated with HA-bound agarose beads to retain IFI27-HA and all its associated proteins, which were then identified by MS. Protein accession number, description, score, mass, coverage, and emPAI index are indicated. **(B)** HEK-293T cells were transiently transfected with pCAGGS-PACT-FLAG in combination with pCAGGS-IFI27-HA or an empty pCAGGS plasmid. At 24 hpt, the cells were left mock-treated or were transfected with poly(I:C) at 3000 ng/ml during 24 h. Cellular extracts were treated with RNAses or left untreated and a Co-IP with HA-bounded agarose beads was performed. PACT and IFI27 were detected by

Western blot employing anti-FLAG (to detect PACT, top panel) and anti-HA (to detect IFI27, bottom panel) antibodies, both in the cellular lysates (Input) and after the Co-IP. Molecular weight is indicated on the right of the panels (in kilodaltons). **(C)** Binding of IFI27 to poly(I:C). Human 293T cells were transiently transfected with the pCAGGS plasmids encoding GFP, IFI27-WT-HA, IFI27-S63L-HA variant, IFI27-V82A-HA variant, and PACT-FLAG, or with an empty plasmid. Pull-down (PD) experiments using poly(I:C)-conjugated agarose beads were performed using cellular extracts. Western blotting using antibodies specific for GFP, the HA tag (to detect IFI27 variants) or the FLAG tag (to detect PACT) was performed to detect protein in the cellular lysates (Input) and after the pull-down (poly(I:C)-PD). Molecular weight markers are indicated (in kilodaltons) on the right. **(D)** HEK-293T cells were transiently transfected with pCAGGS-PACT-FLAG in combination with pCAGGS plasmids encoding IFI27-WT or the IFI27 variants, or an empty pCAGGS plasmid. At 24 hpt, the cells were left mock-treated or were transfected with poly(I:C) at 3000 ng/ml during 24 h. A Co-IP with HA-bounded agarose beads was performed. PACT and IFI27 variants were detected by Western blot employing anti-FLAG (to detect PACT, top panel) and anti-HA (to detect IFI27 variants, bottom panel) antibodies, both in the cellular lysates (Input) and after the Co-IP. Molecular weight is 34 kilodalton for PACT and 12 kilodalton for IFI27. **(E)** HEK-293T cells were transiently transfected with a pCAGGS-IFI27-HA expressing plasmid and 24 hours later, cells were mock-transfected or transfected with 3000 ng/ml of poly(I:C) using polyethylenimine (PEI, polysciences) for 16 hours. Cell lysates were incubated overnight at 4ºC with the PKR specific antibody as well as with protein A-sepharose resin, except in the last Co-IP sample in which the cellular extract expressing IFI27-HA was incubated with the protein A-sepharose resin, without the anti-PKR antibody. Eluates were analysed by Western blot, to detect PKR and IFI27 by employing anti-PKR (to detect endogenous PKR, top panel) and anti-HA (to detect IFI27-HA, bottom panel) both in the cellular lysates (Input) and after the Co-IP. Molecular weight is indicated on the right of the panels (in kilodaltons).

RISC Loading Complex (TARBP2) was found within the interactome with a protein score of 373. This protein is known to interact with PACT and modulate PKR activity [53]. Furthermore, we found some ribosomal proteins, which more likely are involved in protein translation [54,55]. However, due to the lower score for these proteins, in this study we decided to first focus on the likely effect of IFI27 on PKR and PACT functions.

To further confirm the interactions of IFI27 with PKR and PACT, we performed co-immunoprecipitation assays that were analysed by Western blot. HEK-293T cells were transfected with a pCAGGS-IFI27-HA plasmid together with a plasmid expressing PACT fused to a FLAG tag (pCAGGS-PACT-FLAG), and then a co-immunoprecipitation (Co-IP) of IFI27-HA binding proteins was performed, followed by a Western blot analysis (Fig 1B). We were able to confirm the interaction of IFI27 with PACT by this method (Fig 1B). Furthermore, we found that in poly(I:C)-transfected cells, the amount of IFI27 protein co-immunoprecipitated with PACT was clearly decreased, to undetectable levels, in the cell extracts previously treated with a cocktail of RNAses (Fig 1B). Moreover, whereas the amount of PACT in the cellular extracts was similar, the amount of PACT bound to IFI27 was clearly decreased in the cells non-transfected with poly(I:C), compared to the cells transfected with poly(I:C) (Fig 1B), indicating that poly(I:C) favours the interaction of IFI27 with PACT. Nevertheless, a weak interaction of IFI27 with PACT can be detected even in mock-treated cells, suggesting that endogenous RNAs may be mediating this interaction. These results strongly suggest that the interaction of IFI27 and PACT is mediated by dsRNAs, which could be explained by the binding of IFI27 [43] and PACT [26,32] to dsRNAs.

To analyze whether IFI27 interacts with PACT also in infected cells, and to analyze whether the co-immunoprecipitation of IFI27 and PACT also occurs after immunoprecipitating PACT, instead of immunoprecipitating IFI27 as in the previous experiment, the cells were transfected with pCAGGS-IFI27-HA and pCAGGS-PACT-FLAG plasmids and then, the cells were infected with SARS-CoV-2. SARS-CoV-2 infection and the overexpression of PACT and IFI27 were confirmed in the cellular extracts by using antibodies specific for the viral nucleocapsid protein, FLAG (to detect PACT-FLAG), and HA (to detect IFI27-HA) (S1A Fig). Then, a co-immunoprecipitation (Co-IP) of PACT-FLAG binding proteins was performed, followed by a Western blot analysis (S1B Fig). Noteworthy, we were able to confirm the interaction of IFI27 with PACT in SARS-CoV-2-infected cells, and using a FLAG antibody for the immunoprecipitation (S1B Fig), further supporting an interaction of IFI27 with PACT.

According to bioinformatic predictions using RNAbindRPlus, a method that combines machine learning and sequence homology-based approaches [56], IFI27 contains 13 amino acids which are predicted to bind RNA, based on score predictions higher than 0.5. These IFI27 amino acids predicted to bind RNA and their predicted scores are as follows: amino acid 60 (0.68), 61 (0.61), 62 (0.60), 63 (0.53), 64 (0.70), 65 (0.69), 68 (0.64), 69 (0.65), 82 (0.53), 83 (0.58), 84 (0.60), 85 (0.61) and 86 (0.62). In contrast, the predicted scores for the other residues were below 0.1

for 90% of them, and below 0.41 for all of them. According to this data and to further confirm that IFI27 binds PACT in an RNA-dependent manner, we generated two IFI27 variants encoding point mutations in two of these amino acids predicted to bind to RNAs (i.e., S63L and V82A). First, we analyzed whether these IFI27 variants bind to poly(I:C). To this end, cells were transfected with the pCAGGS plasmids expressing IFI27-WT, IFI27-S63L variant, and IFI27-V82A variant fused to HA tags, with a pCAGGS plasmid expressing GFP, as a negative control, or with the plasmid expressing PACT fused to a FLAG tag (pCAGGS-PACT-FLAG), as a positive control of a dsRNA-binding protein [26]. Then, the expression of IFI27 variants, GFP, and PACT was analyzed by Western blot (Fig 1C), and cellular lysates were exposed to agarose beads conjugated to either poly(I:C), an analog of dsRNA (Fig 1C), or poly(C) as control. Remarkably, IFI27 bound the poly(I:C)-conjugated agarose beads, but not to the control poly(C)-conjugated agarose beads, as previously shown by our group [43]. As expected, GFP, did not bind poly(I:C)-agarose beads, and PACT and IFI27 bound poly(I:C)-agarose beads (Fig 1C) [43]. Interestingly, the binding of the IFI27 variants S63L and V82A to poly(I:C)-agarose beads was severely impaired or was abolished, respectively, strongly suggesting that aminoacid changes in positions 63, and 82 affect the binding of IFI27 to poly(I:C) (Fig 1C), as suggested by the bioinformatic predictions.

To analyze whether the IFI27 variants S63L and V82A, which are affected in their binding to poly(I:C), are also affected in their binding to PACT, HEK-293T cells were transfected with the pCAGGS plasmids expressing IFI27-WT, IFI27-S63L variant, and IFI27-V82A variant fused to HA tags, or with the plasmid pCAGGS-PACT-FLAG. Then, the cells were transfected with poly(I:C), and then a co-immunoprecipitation (Co-IP) of IFI27-HA binding proteins was performed, followed by a Western blot analysis (Fig 1D). As previously shown (Fig 1B), the IFI27-WT-HA variant was co-immunoprecipitated with PACT (Fig 1D). However, the IFI27 variants S63L and V82A, which are defective in dsRNA binding, did not interact with PACT (Fig 1D), further indicating that the binding of IFI27 to PACT is mediated by IFI27 binding to dsRNAs. In addition, these data support that the binding of IFI27-WT protein to PACT is specific and is not dependent on the HA tag as these two variants encode only one single amino acid mutation with respect to the IFI27-WT, and the same HA tag as the IFI27-WT protein, and these variants do not bind PACT.

On the other hand, to confirm the interaction between IFI27 and PKR, HEK-293T cells were transfected with a pCAGGS-IFI27-HA plasmid, and then, the cells were either left mock-transfected or transfected with poly(I:C). A specific PKR antibody was bound to protein A-sepharose resin, and the cellular extracts overexpressing IFI27-HA were used to test the binding of IFI27-HA to endogenous PKR. By this method, we were able to confirm the interaction between IFI27 and PKR by a Western blot analysis (Fig 1E), as IFI27 only remained attached to the protein A-resin when the PKR antibody was added. In addition, the results indicated that the interaction of PKR with IFI27 was stronger in the cells transfected with poly(I:C), compared to mock-transfected cells.

To study whether IFI27 colocalizes intracellularly with PACT and/or PKR proteins, HEK-293T cells were transiently transfected with pCAGGS-IFI27-HA alone or together with a plasmid expressing PACT fused to a FLAG tag (pCAGGS-PACT-FLAG). 24 hours after transfection, cells were left mock-transfected or the cells were transfected with poly(I:C) during 24 hours. Then, cells were fixed with formaldehyde and an immunofluorescence was performed in order to detect the intracellular colocalization of IFI27-HA and endogenous PKR or PACT-FLAG (Fig 2A). We were able to observe a partial co-localization of IFI27 and PACT, and of IFI27 and PKR in the cytoplasm. The colocalization area of both fluorophores (Alexa-488 for PACT-FLAG or PKR and Alexa-594 for IFI27-HA) was analyzed and divided by the total cell area (at least n = 29 cells for each condition) obtaining a ratio that was transformed into percentage of colocalization. PACT and IFI27 colocalized in 8.27% of the cell area under mock conditions, which increased to 13.6% after poly(I:C) treatment. Similarly, PKR and IFI27 colocalized in 8.38% of the cell area under mock conditions, a percentage increasing to 13.7% of colocalization after poly(I:C) treatment (Fig 2A and 2B). These results further support the interaction of IFI27 with PACT and PKR from the MS and Co-IP experiments, and, suggest that poly(I:C) favours the interaction of IFI27 with PACT and PKR, supporting the results shown in Fig 1B and 1E.

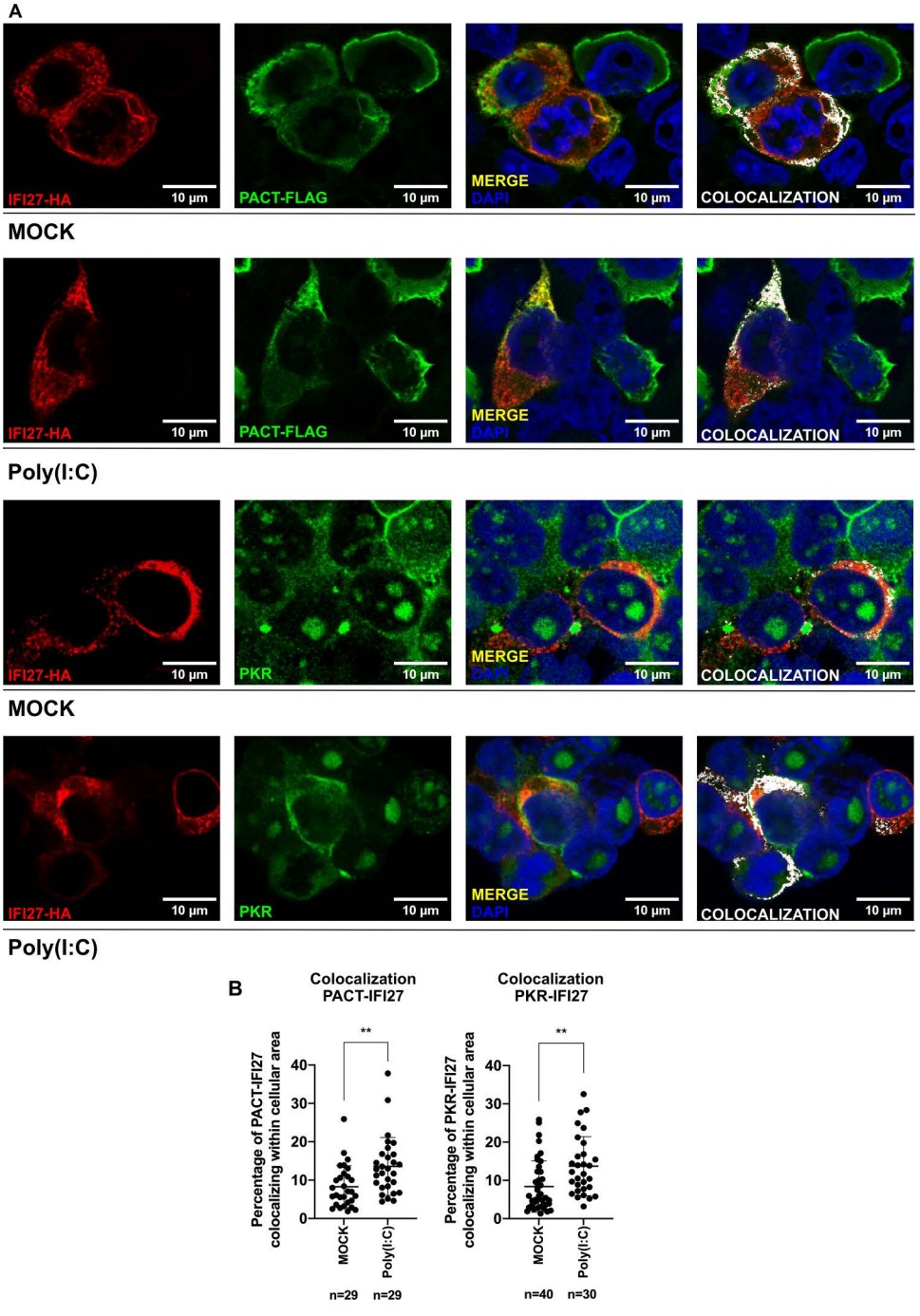

**Fig 2. IFI27 partially colocalizes with PKR and PACT. (A)** HEK-293T cells were transfected with pCAGGS-IFI27-HA together with pCAGGS-PACT-FLAG (for PACT-IFI27 condition) or with pCAGGS-IFI27-HA only (for PKR-IFI27 condition), and then, the cells were either mock-transfected or poly(I:C)-transfected at 3000 ng/ml for 24 hours. After transfection, cells were fixed with formaldehyde and an immunofluorescence was performed to detect IFI27, PACT or endogenous PKR, using anti-HA, anti-FLAG, and anti-PKR antibodies, respectively. Cytoplasm co-localization was analysed and indicated in yellow (merge) in the third picture and white (co-localization) in the fourth picture. **(B)** The level of colocalization between PACT or PKR and IFI27 was analysed by dividing the area where PACT/PKR signal (Alexa Fluor 488, green) and IFI27 signal (Alexa Fluor 594, red) were colocalizing between the total cell area. This ratio was transformed into percentages that are represented on the Y-axis. For PACT-IFI27 mock-transfected and poly(I:C)-transfected conditions, 29 cells were analysed in each case. For PKR-IFI27 mock-transfected and poly(I:C)-transfected conditions, 40 and 30 cells, respectively, were analysed. Each cell result for each condition is represented with dots, and data is represented as the means and standard

deviations of the different measures. ns (non-significative) p > 0.05, *p < 0.05, **p < 0.01, ***p < 0.001, **** p < 0.0001 (for comparisons using unpaired two-tailed Student's t test). Scale bar: 10 µm. These measures were performed in two independent experiments.

## IFI27 overexpression leads to a stronger PKR activation

Upon binding of dsRNAs or RNAs containing duplex regions produced during viral infections, PKR undergoes autophosphorylation, inducing a transition to an active kinase domain [11,16,17]. In addition, a non-canonical PKR activation by endogenous RNAs, such as nuclear and mitochondrial RNAs [57,58], and microRNAs [59] has been described. Activated PKR is characterised by its autophosphorylation on T446 [16]. Activated PKR will then phosphorylate the Ser52 of eIF2α [60], which will result on a general reduction of *de novo* protein synthesis. Not only RNA will participate on the activation of PKR, as there are several protein partners of PKR that will regulate this process, such as PACT. PACT interacts with PKR, facilitating PKR phosphorylation and kinase activity, and promoting host shut-off of protein synthesis [34].

As IFI27 can interact with PACT and PKR proteins (Fig 1), the possible function of IFI27 as a modulator of PKR function was assessed. To this end, we transfected HEK-293T-hACE2 cells with either a pCAGGS-IFI27-HA plasmid or a pCAGGS empty plasmid, in combination with a pRL plasmid expressing Renilla Luciferase (RLuc, pRL-RLuc) under the control of human polymerase II promoter (Fig 3A). This system provides a quantification by luminescence of the total protein synthesis within cells, as the more RLuc is translated, the more luminescence will be detected. At 24 hours post-transfection (hpt), RLuc luminescence signal was analysed (Fig 3B) and IFI27 overexpression was confirmed by Western blot, using an anti-HA specific antibody (Fig 3C and 3F). Interestingly, IFI27 overexpression, led to a 2.7-fold reduction of RLuc luminescence compared to the control cells transfected with the empty plasmid (Fig 3B), even in mock-infected cells. This could be due to a non-canonical PKR activation by endogenous RNAs, in mock-infected and transfected-cells, as it has been published that endogenous RNAs, such as nuclear and mitochondrial RNAs [57,58], and microRNAs [59] can activate PKR.

IFI27 expression is induced after SARS-CoV-2 infection [43]. In order to compare the effect of IFI27 overexpression in the context of viral infections, we transfected HEK-293T-hACE2 cells with either a pCAGGS-IFI27-HA plasmid or a pCAGGS empty plasmid, in combination with the pRL-RLuc plasmid, and 24 hpt, cells were infected with two different, unrelated, RNA viruses, during two different times post-infection, such SARS-CoV-2 (MOI 1), during 6 or 24 hours, or with VSV (MOI 0.1) during 8 h or 24 h (Fig 3A). Viral titers in cell culture supernatants were determined by plaque lysis formation assays, to confirm that the cells were infected with SARS-CoV-2 (24 hours post infection (hpi)) and VSV (8 and 24 hpi) (S2 Fig). After infections, cells were harvested, and cellular extracts were obtained. IFI27 overexpression was confirmed by Western blot, using an anti-HA specific antibody (Fig 3C and 3F). Interestingly, IFI27 overexpression led to a 1.75- and 2.9-fold reduction of RLuc signal compared to the control cells transfected with the empty plasmid, at 6 and 24 h after SARS-CoV-2 infection, respectively (Fig 3B), and 14.7- and 29.3-fold reduction at 8 and 24 h after VSV infection, respectively (Fig 3E). Thus, IFI27 overexpression led to a lower RLuc protein expression in all conditions analysed.

To confirm that this lower RLuc translation levels were due to an increase of PKR activity, protein levels of Ser52 phosphorylated eIF2α (eIF2α-P) and T446 phosphorylated PKR (PKR-P) were measured by Western blot and compared to the total eIF2α and PKR levels (Fig 3C and 3F). At 6 h after SARS-CoV-2 infection, eIF2α-P levels increased 2-fold, similarly in cells transfected with the empty and IFI27 plasmids, compared to mock conditions (Fig 3C and 3D). At 24 hpi, eIF2α-P levels increased 2-fold on empty plasmid-transfected cells, but interestingly, this increase was higher (3.9-fold) on IFI27 overexpressing cells, compared to the empty control mock-infected cells at 24 hpi (Fig 3C and 3D). PKR-P levels remained similar between empty and IFI27 plasmid-transfected cells in mock-infected cells 6 h, but they were increased by IFI27 overexpression on mock-infected cells at 24 h compared to empty transfected conditions (1.8-fold) (Fig 3C and 3D). At 6 h after SARS-CoV-2 infection, PKR-P levels were increased 7.4- and 9.6-fold in empty and IFI27 transfection conditions, respectively. However, at 24 hpi, PKR phosphorylation was decreased by SARS-CoV-2 infection by 1.2- and

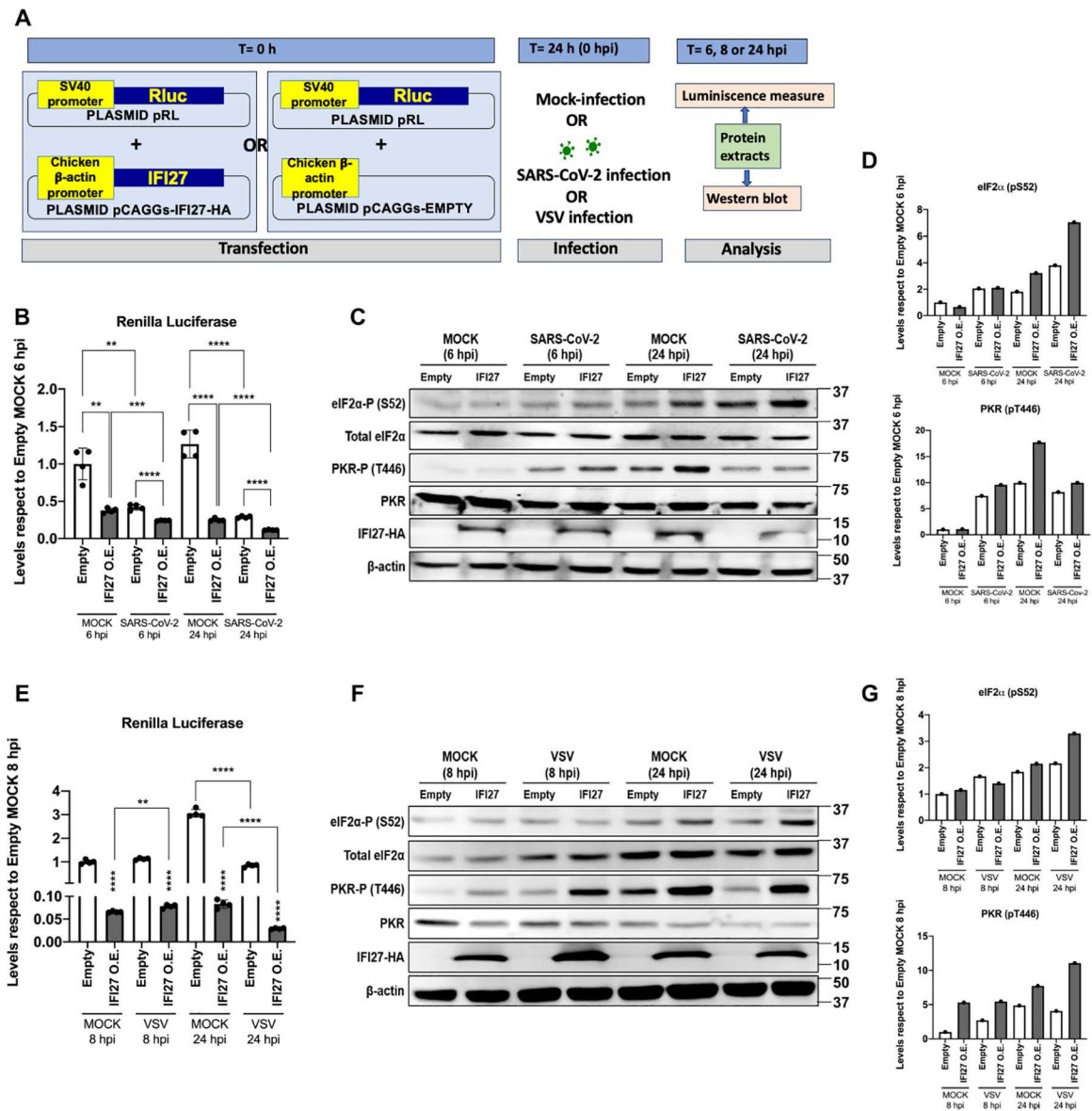

**Fig 3. IFI27 overexpression leads to a stronger PKR activation. (A)** Schema of the experimental timeline. **(B, C, D)** HEK-293T-hACE2 cells were transiently transfected with a pCAGGS-IFI27-HA plasmid (IFI27 O.E.) or an emtpy pCAGGS plasmid (Empty) together with a pRL plasmid expressing RLuc luciferase, and 24h later, transfected cells were left mock-infected or infected with SARS-CoV-2 at a multiplicity of infection (MOI) of 1. At 6 and 24 hours post-infection (hpi), protein extracts were obtained and the levels of RLuc luminescence were measured. Individual data points correspond to independent measures from different technical samples **(B)**. All luminescence levels were compared respect to mock-infected cells transfected with the empty plasmid **(B)**. **(C)** These protein extracts were also used to measure protein levels of eIF2α-P (S52), total eIF2α, PKR-P (T446), total PKR, IFI27-HA, and β-actin (loading control) by Western blot employing their respective specific antibodies. Molecular weight is indicated on the right of the panels (in kilodaltons). Western blots were quantified by densitometry using ImageJ software. The amount of eIF2α-P (S52) was normalised to the amount of total eIF2α and β-actin. The amount of PKR-P (T446) was normalised to the amounts of total PKR and β-actin (results of quantification showed in bars to the right of immunoblots **(D)**), with levels relative to empty plasmid-transfected, mock-infected cells at 6 hpi. **(E, F, G)** HEK-293T cells were transiently transfected with a pCAGGS-IFI27-HA plasmid (IFI27 O.E.) or an emtpy pCAGGS plasmid (Empty) together with a pRL plasmid expressing RLuc luciferase, and 24h later, transfected cells were infected with VSV at a MOI 0.1. At 8 or 24 hpi, protein extracts were obtained. **(E)** The levels of RLuc luminescence were measured. All luminescence levels were compared respect to cells transfected with the empty plasmid and mock-infected during 8 h. Individual data points correspond to independent measures from different technical samples. **(F)**. These protein extracts were also used to measure protein levels of eIF2α-P (S52), total eIF2α, PKR-P (T446), total PKR, IFI27-HA, and β-actin by Western blot employing their respective antibodies. Molecular weight is indicated on the right of the panels (in kilodaltons). Western blots were quantified by densitometry using ImageJ

software. The amount of eIF2α-P (S52) was normalised to the amounts of total eIF2α and β-actin. The amount of PKR-P (T446) was normalised to the amounts of total PKR and β-actin (results of quantification shown in bars to the right of immunoblots, with levels relative to cells transfected with the empty plasmid and mock-infected during 8 h (G)). Data is represented as the means and standard deviations of the differentmeasures (Individual data points correspond to independent measures from different technical samples). p > 0.05, *p < 0.05, **p < 0.01, ***p < 0.001, **** p < 0.0001 (for comparisons using unpaired two-tailed Student's t test in **B** and **E)**. The asterisks above each bar represent the comparison vs. the empty control at each time and condition. These measures were performed in at least two independent experiments.

1.7-fold in empty and IFI27 transfection conditions, respectively, compared to mock-infected cells (Fig 3C and 3D). This could be due to 24 hours post-infection (hpi) being a relatively late time after infection and to the fact that it has been previously described that SARS-CoV-2 protein N is able to antagonize PKR activation [61], as well as the nsp15 of coronaviruses could have an important function on antagonizing PKR-eIF2α antiviral pathway to ensure an efficient replication [24,25,62]. Furthermore, eIF2α-P levels increased 1.5-fold when cells overexpressed IFI27 compared to the control cells transfected with the empty plasmid, at 24 h after VSV infection (Fig 3F and 3G). PKR-P levels behaved similarly, with IFI27 overexpression increasing PKR-P near 2-fold at 8 and 24 h after VSV infection, compared to the control cells transfected with the empty plasmid. These results strongly suggest that IFI27 positively modulates PKR activation, and the subsequent eIF2α phosphorylation.

Since we found that even under mock conditions, transfection of IFI27-encoding plasmid induces a reduction of RLuc production by activating the PKR pathway (Fig 3B and 3E), we wanted to analyse whether there was a dose-dependent effect of protein translation inhibition by using increasing amounts of pCAGGS-IFI27-HA plasmid. To this end, we transfected HEK-293T-hACE2 cells with increasing amounts of either a pCAGGS-IFI27-HA plasmid or a pCAGGS empty plasmid (0.25, 0.5 and 1 ng/µl), in combination with a stable amount of pRL-RLuc plasmid. At 24 hours post-transfection, cells were either mock or VSV infected, at a MOI of 0.1, and 24 hours after infection, cells were harvested, and protein extracts were used to measure RLuc signal (S3A Fig) and IFI27-HA expression was confirmed by Western Blot (S3B and S3C Fig). In mock-infected cells, we observed an increase of IFI27-HA levels of expression with the increasing concentrations of plasmid (S3B and S3D Fig) which correlated with a decrease on RLuc signal (S3A Fig). Similarly, in VSV-infected cells, we observed an increase of IFI27-HA levels from 0.25 ng/µl to 0.5 ng/µl concentrations, and an slightly lower increase from 0.5 ng/µl to 1 ng/µl plasmid concentrations (S3C and S3D Fig). This increase in IFI27 expression levels, negatively correlates with the level of production of RLuc, as we observe a significant decrease from 0.25 ng/µl to 0.5 ng/µl conditions and from 0.5 ng/µl to 1 ng/µl conditions. In every concentration and infection-condition, IFI27 overexpression leads to a lower level of luminescence compared to the empty control. This correlation between the levels of IFI27-HA and RLuc, further show the ability of IFI27 to induce PKR and eIF2α phosphorylation, and further promote host shut-off after viral infections, being this effect dose-dependent.

### IFI27 knock-out and knock-down results in a weaker PKR activation and higher *de novo* protein synthesis

SARS-CoV-2 and VSV are unrelated viruses, which activate the PKR kinase activity in order to reduce protein synthesis [63–65]. To confirm that IFI27 positively modulates PKR activity in a system different to overexpression, we used parental (WT) and IFI27 KO human A549 cells overexpressing hACE2 (IFI27 KO A549-hACE2), previously generated [43]. We sequenced the IFI27 KO cells, showing a 32 nt deletion within the first nucleotides of the open reading frame and confirmed that the IFI27 KO cells did not expressed the IFI27 protein by Western blot, as previously published [43]. Then, the WT and IFI27 KO cell lines were transfected with RLuc expressing plasmid. 24 hours later, the cells were transfected with increasing concentrations of poly(I:C), a synthetic dsRNA that simulates viral dsRNA and promotes PKR activation [66]. Both in mock-transfected, and poly(I:C)-transfected cells, the RLuc luminescence levels were 2.2 to 2.5-fold higher in IFI27 KO A549-hACE2 cells compared to parental cells (Fig 4A). Furthermore, the transfection of cells with increasing concentrations of poly(I:C) slightly reduced RLuc luminescence both in WT and KO cells strongly suggesting that poly(I:C)

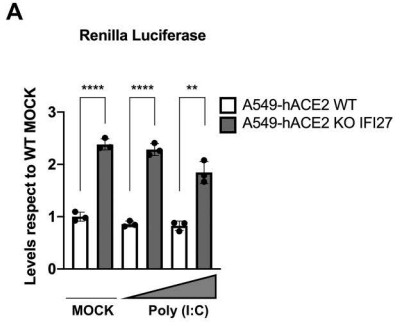

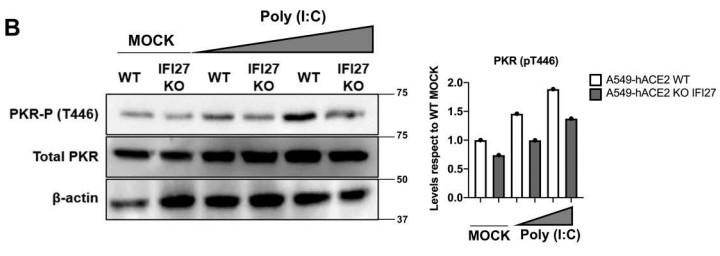

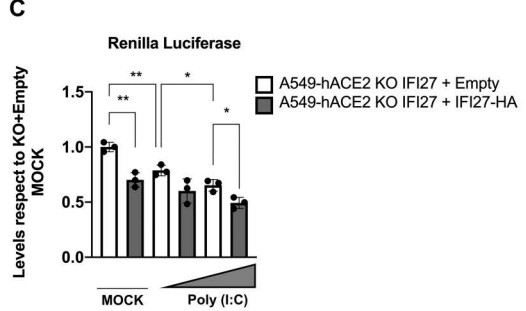

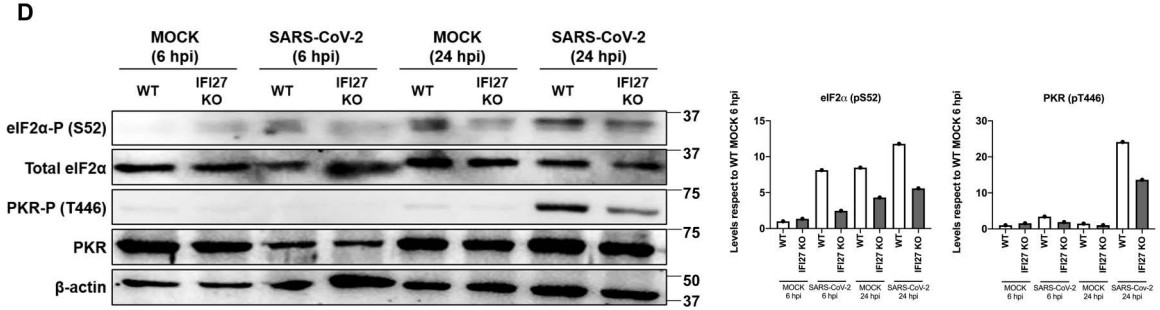

**Fig 4. IFI27 KO impairs PKR activation after poly(I:C) treatment. (A, B)** WT or IFI27 KO A549-hACE2 cells were transiently transfected with a pRL plasmid expressing RLuc luciferase (pRL-Rluc), and 24h later, the cells were mock-transfected or transfected with increasing concentrations of poly(I:C) (50 or 250 ng/ml) for 7 **h. (A)** Protein extracts were obtained and the levels of RLuc luminescence were measured. All luminescence levels were compared with respect to mock-transfected, A549-hACE2 WT cells. Individual data points correspond to independent measures from different technical samples. ns (non-significant) $p > 0.05$, *$p < 0.05$, **$p < 0.01$, ***$p < 0.001$, **** $p < 0.0001$ (for comparisons using unpaired two-tailed Student's t test in (**A**). **(B)** Protein extracts obtained by lysis were also used to measure protein levels of PKR-P (T446), total PKR and β-actin (loading control) by Western blot, employing their respective specific antibodies. Molecular weight is indicated on the right of the panels (in kilodaltons). Western blots were quantified by densitometry using ImageJ software. The amount of PKR-P (T446) was normalised by total PKR and β-actin (results of quantification showed in bars below immunoblots, with levels relative to mock-transfected A549-hACE2 WT cells). **(C)** A549-hACE2 IFI27 KO cells stably transfected with an empty plasmid (KO+emtpy), and IFI27 KO cells stably expressing IFI27-HA (KO + IFI27), were transiently transfected with the plasmid pRL-Rluc and 24h later, the cells were mock-transfected or transfected with increasing concentrations of poly(I:C) (50 or 250 ng/ml) for 8 h. Protein extracts were obtained and the levels of RLuc luminescence were measured. All luminescence levels were compared with respect to mock-transfected, A549-hACE2 IFI27 KO cells stably transfected with an empty plasmid. Individual data points correspond to independent measures from different technical samples. **(D)** WT or IFI27 KO A549-hACE2 cells were infected with SARS-CoV-2 (MOI 0.5). Protein extracts were obtained and used to measure protein levels of eIF2α-P (S52), total eIF2α, PKR-P (T446), total PKR and β-actin (loading control) by Western blot employing their respective antibodies. Molecular weight is indicated on the right of the panels (in kilodaltons). Western blots were quantified by densitometry using ImageJ software. The amount of eIF2α-P (S52) was normalised to the amounts of total eIF2α and β-actin. The amount of PKR-P (T446) was normalised to the amounts of total PKR and β-actin. Results of quantification are shown in bars to the right of immunoblots, with levels relative to mock-infected A549-hACE2 WT cells at 6 hpi. These measures were performed in at least two independent experiments.

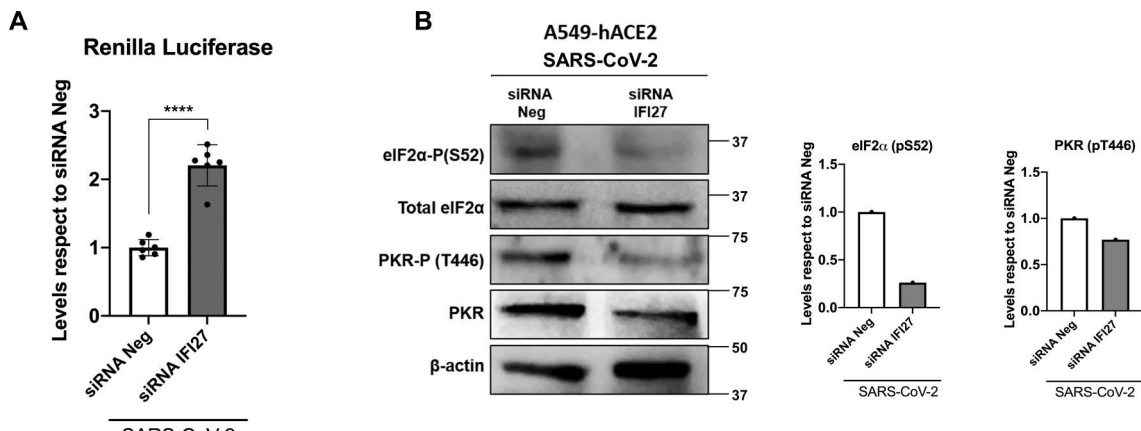

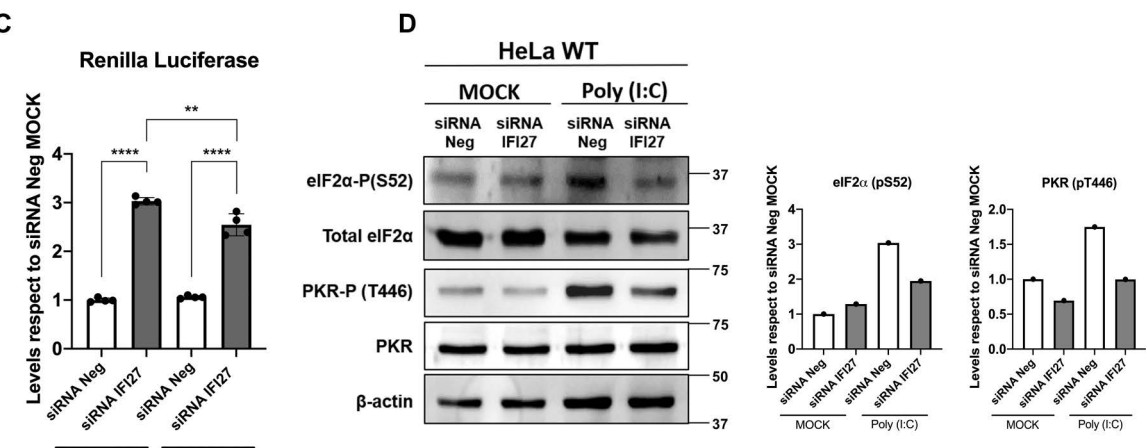

**Fig 5. IFI27 knock-down impairs PKR activation after SARS-CoV-2 infection and after poly(I:C) transfection. (A, B)** IFI27 knock-down was performed on A549-hACE2 WT cells. Cells were transfected twice, with a 24-hour gap between each transfection, either with a negative control non-targeting siRNA (siRNA Neg) or with an IFI27 siRNA (siRNA IFI27). 24 hours after the second siRNA transfection, cells were transfected with a pRL plasmid expressing RLuc luciferase, and 6 h later, cells were mock or SARS-CoV-2-infected for 24 hours (MOI 1). Protein extracts were obtained by lysis. **(A)** The levels of RLuc luminescence were measured (luminescence levels were compared respect to cells transfected with the control siRNA). Individual data points correspond to independent measures from different technical samples. **(B)** Protein extracts were used to measure protein levels of eIF2α-P (S52), total eIF2α, PKR-P (T446), total PKR and β-actin by Western blot employing their respective antibody. Molecular weight is indicated on the right of the panels (in kilodaltons). Western blots were quantified by densitometry using ImageJ software. The amount of eIF2α-P (S52) was normalised by total eIF2α and β-actin. The amount of PKR-P (T446) was normalised by total PKR and β-actin (results of quantification showed in bars at the right side of immunoblots, with levels relative to cells transfected with the control siRNA). **(C, D)** IFI27 knock-down was performed on HeLa WT cells. Cells were transfected twice, with a 24-hour gap between each transfection, either with a negative control non-targeting siRNA (siRNA Neg) or with an IFI27 siRNA (siRNA IFI27). 24 hours after the second siRNA transfection, cells were transfected with a pRL plasmid expressing RLuc luciferase, and 6 h later, cells were mock or poly(I:C)-transfected at 2 µg/ml for 16 hours. Protein extracts were obtained by lysis and the levels of RLuc luminescence were measured **(C)** (luminescence levels were compared respect to cells transfected with the control siRNA and mock-transfected, individual data points correspond to independent measures from different technical samples) or **(D)** used to measure protein levels of eIF2α-P (S52), total eIF2α, PKR-P (T446), total PKR and β-actin by Western blot employing their respective antibody. Molecular weight is indicated on the right of the panels (in kilodaltons). Western blots were quantified by densitometry using ImageJ software. The amount of eIF2α-P (S52) was normalised by total eIF2α and β-actin. The amount of PKR-P (T446) was normalised by total PKR and β-actin (results of quantification showed in bars at the right side of immunoblots, with levels relative to cells transfected with the control siRNA and non-transfected with poly(I:C)). Data is represented as the mean and standard deviations of the different measures. ns (non-significant) p > 0.05, *p < 0.05, **p < 0.01, ***p < 0.001, **** p < 0.0001 (for comparisons using unpaired two-tailed Student's t test in **A** and **C**). These measures were performed in two independent experiments.

transfection activates PKR, reducing protein synthesis. PKR activation was further confirmed by Western blot detection of PKR-P. PKR-P levels in comparison to total PKR were augmented with increasing concentrations of poly(I:C) compared to mock conditions both in WT and IFI27 KO cells (Fig 4B). Therefore, higher RLuc luminescence correlated with lower PKR-P levels, as previously described [67]. Interestingly, PKR-P levels were near 1.5-fold lower in IFI27 KO cells compared to WT cells both in mock-transfected and poly(I:C)-transfected cells (Fig 4B), showing that IFI27 expression positively modulates PKR activation.

To analyze whether the overexpression of IFI27 in IFI27 KO cells, rescues the phenotype, A549-hACE2 IFI27 KO cells stably transfected with an empty plasmid, and IFI27 KO cells stably expressing IFI27-HA by means of a linearized plasmid transfection, previously generated and described by us [44] were used. After confirming IFI27 overexpression [44], these cell lines were transfected with the plasmid expressing Rluc, and then, the cells were treated with different concentrations of poly(I:C) for 8 hours. The results show that as expected given our previous results, the overexpression of IFI27 decreases Rluc expression in IFI27 KO cells, rescuing the phenotype (Fig 4C).

Plasmid transfections affect the interferon and PKR-dependent pathways, as previously shown [26,68]. However, and to avoid transfection bias, transfections with the same amount of an empty plasmid, and lipofectamine, as control, are included in all of our experiments. In addition, to analyze the effect of IFI27 on PKR activation in the absence of transfection, we analyzed eIF2α and PKR phosphorylation in SARS-CoV-2-infected cells, at two time points, in WT and IFI27 KO cells (Fig 4D), in the absence of any transfection. The results show higher levels of eIF2α and PKR phosphorylation in WT cells, compared to IFI27 KO cells (Fig 4D), further indicating that IFI27 positively affects PKR activation after RNA virus infections, even in the absence of cell transfections.

To further confirm that IFI27 down-regulation could impair PKR activation after a viral infection, A549-hACE2 cells were transfected twice, with an siRNA specific for IFI27 or a non-targeting siRNA as a control. At 24 h after the second round of siRNA transfection, cells were transfected with RLuc expressing plasmid and 6 hours later infected with SARS-CoV-2 for 24 hours. Both RLuc luminescence (Fig 5A) and the band intensity of eIF2α-P and PKR-P on Western blot (Fig 5B) were analysed. IFI27 knockdown was confirmed by qRT-PCR, with a strong reduction of IFI27 mRNA levels after mock or poly(I:C) treatments and SARS-CoV-2 infection respect to control (siRNA Neg) (S4 Fig). IFI27 knock-down led to a 2.2-fold increase of RLuc luminescence compared to the control siRNA (Fig 4A), which correlated with a lower band intensity of both normalized eIF2α-P and PKR-P compared to the control siRNA (Fig 5A and 5B). These results show an impaired PKR activation after IFI27 knock-down.

IFI27 knock-down was performed also on HeLa cell line, to corroborate that the observed differences on PKR function were not cell-specific. HeLa WT cells were transfected twice with an IFI27 siRNA or a non-targeting siRNA, as control. At 1 day post-second round of siRNA transfection, cells were transfected with RLuc expressing plasmid and 6 hours later, the cells were transfected with poly(I:C) for 16 hours. Both RLuc luminescence (Fig 5C) and the band intensities of eIF2α-P and PKR-P on Western blot (Fig 5D) were analysed. Similar to the results obtained on A549-hACE2 cells, IFI27 knock-down led to a near 3-fold increase of RLuc luminescence both on mock-transfected and poly(I:C)-transfected conditions compared to the control siRNA (Fig 5C). These results correlated with the weaker band intensity of both normalized eIF2α-P and PKR-P in the cells knocked-down for IFI27, compared to the control cells transfected with the non-targeting siRNA (Fig 5D). These results show, again, that IFI27 downregulation leads to a weaker PKR activation also in HeLa cells.

## PACT is required for the activation of PKR by IFI27 protein

PACT is a dsRNA binding protein that can activate PKR in a dsRNA-independent manner [26]. PACT binds PKR at several domains: the two dsRNA-binding domains (dsRBDs) and the N-terminal part of the kinase domain [69,70]. When PACT binds PKR, it shifts the equilibrium to the active form of PKR, in which the dsRBDs are exposed and able to recognise and bind viral dsRNAs [70,71]. As both PACT and PKR were present on the protein interactome of IFI27 (Fig 1A), and knowing the importance of PACT on PKR activation, we hypothesized that IFI27 ability of inducing the activation of

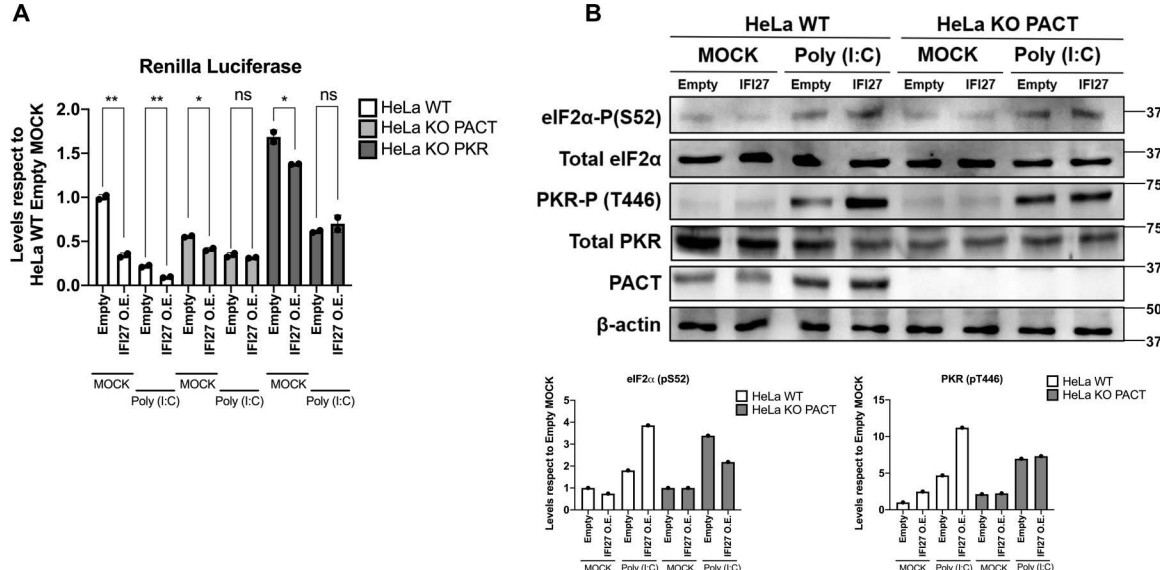

**Fig 6. PACT absence impairs IFI27-mediated PKR activation (A)** HeLa WT, PACT KO and PKR KO cells were transiently transfected with a pCAGGS-IFI27-HA plasmid (IFI27 O.E.) or an emtpy pCAGGS plasmid (Empty) together with a pRL plasmid expressing RLuc luciferase, and 24h later, cells were mock-transfected or transfected with poly(I:C) at 2 µg/ml for 7 hours. At 24 hours post-treatment, protein extracts were obtained by lysis and the levels of RLuc luminescence were measured (A). All luminescence levels were compared with respect to the levels in HeLa WT cells transfected with the empty plasmid. Individual data points correspond to independent measures from different technical samples. **(B)** Same protein extracts from HeLa WT and HeLa PACT KO cells, either mock-transfected or transfected with poly(I:C) were employed to measure the protein levels of eIF2α-P (S52), total eIF2α, PKR-P (T446), total PKR and β-actin by Western blot employing their respective antibodies. Molecular weight is indicated on the right of the panels (in kilodaltons). Western blots were quantified by densitometry using ImageJ software. The amount of eIF2α-P (S52) was normalised by total eIF2α and β-actin. The amount of PKR-P (T446) was normalised by total PKR and β-actin (results of quantification showed in bars below the immunoblots, with levels relative to HeLa WT cells transfected with the empty plasmid, under mock conditions, non-transfected with poly(I:C)). Data is represented as the mean and standard deviations of the different measures, being the data representative of two independent experiments. ns (non-significant) $p > 0.05$, *$p < 0.05$, **$p < 0.01$, ***$p < 0.001$, **** $p < 0.0001$ (for comparisons using unpaired two-tailed Student's t test in A).

PKR could be facilitated by PACT. To test this hypothesis, we used HeLa cell lines knocked-out (KO) for the genes PACT or PKR, previously generated [17]. These KO cells were tested by Western blot of the respective knocked-out gene (Fig 6B for PACT KO cells, and S5 Fig for PKR KO cells). Then, cells were transfected with either a pCAGGS-IFI27-HA plasmid or a pCAGGS empty plasmid, in combination with a pRL plasmid expressing RLuc during 24 hours. After, cells were mock-transfected or poly(I:C)-transfected (Fig 6A) during 7 hours, and RLuc luminescence was analysed. In PKR KO cells the levels of Rluc expression were higher than in the WT cells (Fig 6A), as expected, as PKR deficiency leads to an increase on general protein expression and thus, stronger RLuc luminescence [67]. As expected, in HeLa WT cells, the overexpression of IFI27 led to a decrease on RLuc luminescence (3-fold) both on mock and poly(I:C)-transfected conditions (Fig 6A), compared to the cells transfected with the empty plasmid. For PACT, and PKR KO cells, we only observed a slight change on luminescence after IFI27 overexpression on mock conditions compared to control cells transfected with the empty plasmid (Fig 6A). Furthermore, after poly(I:C) treatment neither PACT nor PKR KO cells showed significantly decrease of RLuc luminescence after IFI27 expression (Fig 6A). These results strongly indicate that the absence of PACT and PKR also led to an impairment on the ability of IFI27 to reduce luminescence by PKR activation, and that IFI27 overexpression affects protein translation in a PKR- and PACT-dependent manner. Noteworthy, although PACT is an activator of PKR under stress conditions, it has been shown in two preprints that PACT can act as a direct and specific suppressor of PKR against endogenous dsRNA ligands, which can activate PKR in the absence of PACT [72,73]. Therefore, the

decreased protein translation in the PACT KO, mock-treated cells, transfected with the empty plasmid could be due to this PACT function of decreasing PKR activation mediated by endogenous dsRNAs or by plasmid-derived RNAs.

To further confirm that the ability of IFI27 to modulate PKR activity depends on PACT, the band intensities of eIF2α-P and PKR-P on Western blot were analysed under the same treatment conditions on HeLa WT and HeLa PACT KO cells (Fig 6B). In HeLa WT cells, poly(I:C) treatment led to a 1.8-fold increase of normalised eIF2α-P and 4.7-fold increase of normalised PKR-P intensity on the control cells transfected with the empty plasmid (Fig 6B), as expected, as it was previously published that poly(I:C) treatment induces PKR activation [74]. Interestingly, this increase of eIF2α-P and PKR-P in the WT cells transfected with poly(I:C) was even higher when IFI27 was overexpressed, 2.1 and 2.4-fold, respectively, compared to when IFI27 was not overexpressed (Fig 6B). However, in the PACT KO cells, the increase on eIF2α-P and PKR-P intensity after poly(I:C) transfection was similar between the empty-plasmid transfected condition, and IFI27 over-expression condition (Fig 6B). These results correlated with the RLuc luminescence values (Fig 6A), and strongly suggest that the effect of IFI27 overexpression on PKR activation, even in transfected cells, is specific and highly depends on PKR and PACT expression.

## IFI27 overexpression promotes Stress Granules (SGs) formation in a PACT-PKR dependent manner

The translational arrest induced by activated PKR leads to the formation of SGs [75,76]. These structures are cytoplasmic aggregates that include stalled translation initiation complexes composed by 40S ribosomal subunits, translation initiation factors such as eIF3, poly(A)+ mRNAs and RNA binding proteins (RBPs), and additionally, other components such as the nucleator factor Ras GTPase-activating protein-binding protein 1 (G3BP1), T cell intracytoplasmatic antigen 1 (TIA1) and others [76–78]. SGs have also been described to be important platforms for antiviral signalling pathways, as these ribonucleoprotein complexes were found to further regulate PKR activation [79]. Furthermore, PKR was shown to be able to regulate the antiviral innate responses not only by inducing translation stalling, but also by working as a regulator of other components of this response [75]. SGs are a hub of connection between the stress response and the antiviral immune response. From the SGs, PKR can regulate the activation of RIG-I [80], MDA5 [81], MAVS [82], and cGAS-STING [83], as well as that of immune transcription factors such as NF-κB [84], STAT1 [85], STAT3 [86]. Thus, the presence of these SGs is crucial for a proper antiviral response and regulation of signalling pathways within the cell.

As IFI27 can potentiate PKR activation, one of the main inductors of SGs [87], we determined whether IFI27 overexpression could indirectly increase the formation of SGs by positively affecting the activation of PKR. SGs formation after poly(I:C) treatment was confirmed on HeLa WT cells by immunostaining of the SGs with the markers eIF3η (also referred to as EIF3A) [77] and G3BP1 [76]. In addition, these markers (eIF3η and G3BP1) colocalized with PKR granules (Fig 7A), confirming that these three elements are present on poly(I:C)-induced SGs. To analyse the effect of IFI27 on SG assembly, HeLa WT and HeLa PACT KO cells were transfected with either a pCAGGS-IFI27-HA plasmid or a pCAGGS empty plasmid, and 24 hours later, the cells were treated with poly(I:C) to induce the activation of PKR and antiviral pathways. Then, an immunofluorescence for both eIF3η and G3BP1 proteins was performed. The percentage of SGs-containing cells on the total of empty plasmid-transfected HeLa WT and HeLa PACT KO cells was of 25.8 and 26.6% respectively, considering both eIF3η and G3BP1 positive granules (Fig 7B and 7C). Interestingly, IFI27 overexpression induced a strong and significative increment on the percentage of SGs- containing cells on HeLa WT cells up to a 47.4%, which is almost double compared to the % in empty plasmid-transfected WT cells (Fig 7B and 7C). However, this was not the case for HeLa PACT KO cells, as in the IFI27-overexpressing PACT KO cells, the SGs containing cells were the 32.1%, which was not a statistically significant increment (Fig 7B and 7C), compared to the empty plasmid-transfected PACT KO cells.

To further dissect the effect of IFI27 on SG formation in presence or absence of PACT, we transfected both HeLa WT and HeLa PACT KO cells with a pCAGGS-IFI27-HA plasmid and 24 hours later, the cells were treated with poly(I:C) for 7 hours. Then, the cells were fixed and processed to perform an immunofluorescence to detect both IFI27-HA and G3BP1 as a SG marker. Now, we measured the percentage of SGs containing cells only on those cells that presented a positive

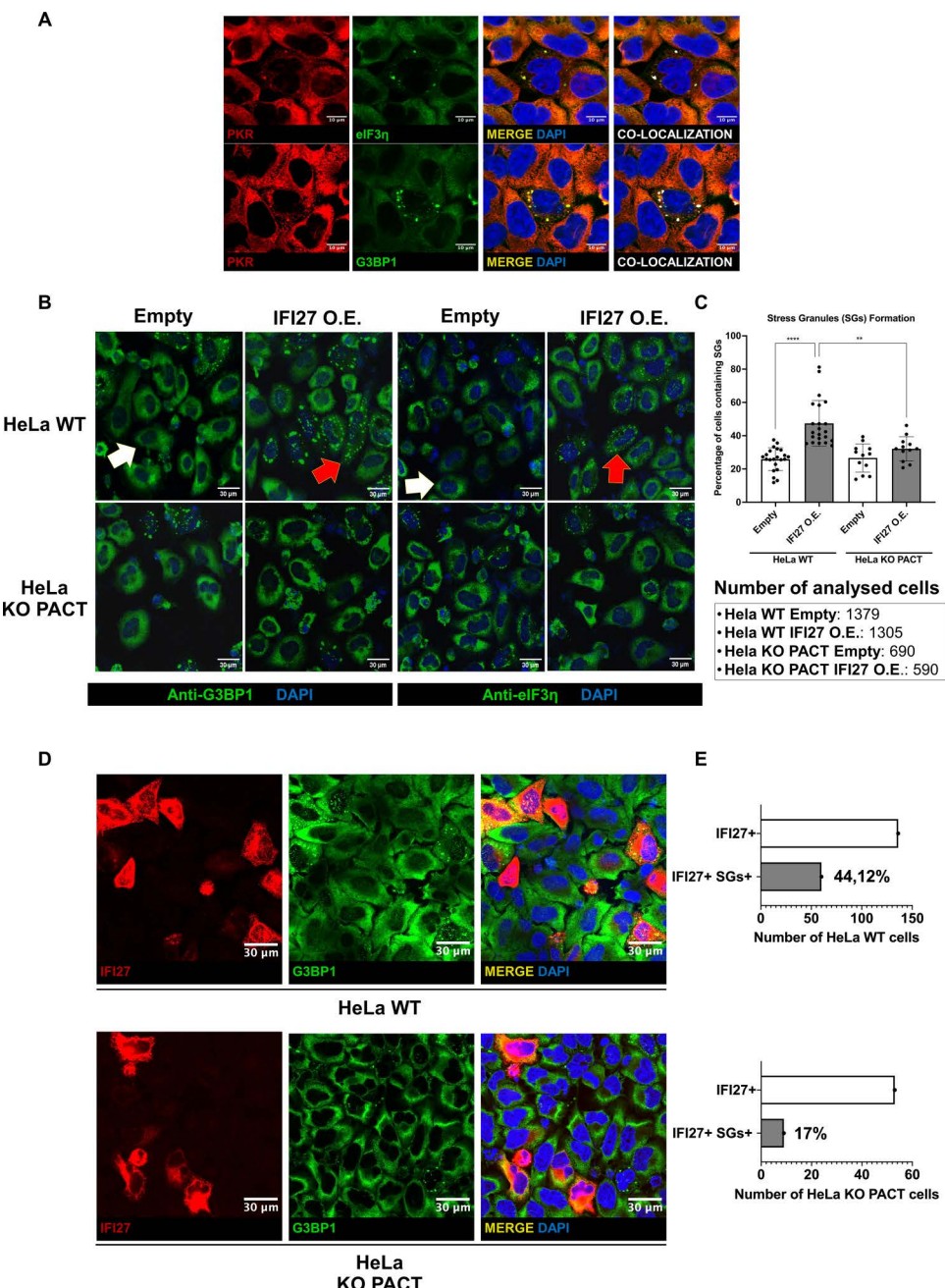

**Fig 7. IFI27 promotes the formation of SGs in a PACT-dependent manner. (A) HeLa WT cells were treated with poly(I:C) at 2 µg/ml for 7 hours and then, cells were fixed with formaldehyde and permeabilized.** PKR and eIF3η or PKR and G3BP1 were labelled with specific antibodies for each protein. PKR is shown in red, eIF3η and G3BP1 are shown in green, and nuclei were stained with DAPI and shown in blue. Areas of co-localization of both proteins appear in yellow in the third picture and in white in the fourth picture. Scale bar, 10 µm. **(B and C)** HeLa WT and HeLa PACT KO cells were transiently transfected with either a pCAGGS-IFI27-HA plasmid or a pCAGGS empty plasmid, and 24 hours later, treated with poly(I:C) at 2 µg/ml for 7 hours. Then, cells were fixed with formaldehyde and permeabilized. **(B and C)** An immunofluorescence was performed for both eIF3η and G3BP1 as markers of SGs. White arrows **(B)** show cells with no presence of SGs, while the red arrows **(B)** showcells with positive presence of SGs. **(C)** Quantification of the percentage of SGs positive cells within the total number of counted cells for each condition is represented. Each dot indicates the percentage of SGs positive cells within a single region, having each region several cells. The total number of counted cells was 1379 and 1305 cells for WT cells transfected with the empty plasmid and the plasmid for IFI27 overexpression, and 690 and 590 cells for PACT KO cells transfected with the empty plasmid and the plasmid for IFI27 overexpression. Scale bar, 30 µm. Data is represented as the mean and standard deviations of the different measures.

ns (non-significant) p > 0.05, *p < 0.05, **p < 0.01, ***p < 0.001, **** p < 0.0001 (for comparisons using unpaired two-tailed Student's t test in **C**). **(D and E)** HeLa WT and HeLa PACT KO cells were transiently transfected with a pCAGGS-IFI27-HA plasmid, and 24 hours later, treated with poly(I:C) at 2 µg/ml for 7 hours. Then, cells were fixed with formaldehyde and permeabilized. IFI27-HA and G3BP1 were labelled with specific antibodies for each protein. **(D)** IFI27-HA is shown in red, G3BP1 is shown in green, and nuclei were stained with DAPI and shown in blue. Areas of co-localization of both proteins appear in yellow in the third picture. Scale bar, 30 µm. **(E)** A quantification of the total IFI27-HA positive cells counted (white bar), and the number of SGs positive cells within the total IFI27-HA positive cells (grey bar) is represented. The number of counted cells is shown on the x-axis, being the counted cells 136 and 60 for WT cells overexpressing IFI27 and WT cells overexpressing IFI27 and containing SGs, respectively, and 53 and 9 for PACT KO cells overexpressing IFI27 and PACT KO cells overexpressing IFI27 and containing SGs, respectively. The percentage IFI27-HA positive cells containing SGs within the total IFI27-HA positive cells is shown next to the grey bar.

signal of IFI27-HA expression. IFI27 overexpression induced the formation of detectable SGs on the 44.12% of HeLa WT cells, while only the 17% of IFI27-HA-positive cells showed the formation of SGs on HeLa PACT KO cells (Fig 7D and 7E). These results correlate with previous results (Fig 5), and confirm that IFI27 induces PKR activation, and thus, SGs formation, and that this ability is dependent on the presence of PACT.

### PKR overexpression induces the formation of SGs, which is potentiated by co-overexpression with IFI27

PKR overexpression leads to an induction of its own activation and the induction of SGs on HeLa PKR KO cells without the need for any other stimulus [17]. Taking into account this knowledge, to further investigate the ability of IFI27 to modulate PKR activity in collaboration with PACT, parental and IFI27 KO human A549-hACE2 cells were transfected with a pCAGGS plasmid expressing PKR-myc or with an empty pCAGGS plasmid, as control, in combination with RLuc expressing plasmid, to determine if the absence of IFI27 impaired PKR activation. RLuc luminescence levels in control cells transfected with the empty plasmid were about 5-fold higher in IFI27 KO cells than in WT cells, showing how the absence of IFI27 leads to less activation of endogenous PKR (Fig 8A). When PKR is overexpressed, RLuc luminiscence levels drop considerably in both WT and IFI27 KO cells, reflecting an increased activation of the PKR pathway. However, in IFI27 KO cells, we again detected a higher luminescence (more than double) than in WT cells, again being a sign of lower PKR activation in the IFI27 KO cells (Fig 8A). To rule out that the differences in Rluc expression between WT and IFI27 KO cells are not due to differences in transfection efficiency between both WT and IFI27 KO cell lines, we have performed an RT-qPCR experiment in which we have analyzed the expression of Rluc at the mRNA level (S6 Fig). These new results indicate that the expression of Rluc mRNAs is the same in both cell lines, indicating that there are no significant differences in the transfection efficiency and in Rluc transcription between WT and IFI27 KO cells.

The next objective was to determine, under conditions of PKR overexpression, the importance of PACT for IFI27-mediated PKR activation. Therefore, HeLa WT, PACT KO, PKR KO and also HeLa KO for both PACT and PKR (DKO) were transfected with either only a PKR-myc pCAGGS (PKR) or a combination of PKR-myc and IFI27-HA pCAGGS plasmids (PKR + IFI27), plus a RLuc expressing plasmid, to determine the level of general translation. After 24 hours of transfection, cells were either mock-transfected or transfected with poly(I:C) for 7 hours, to further potentiate PKR activation. On mock-transfected conditions, the overexpression of PKR and IFI27 decreased RLuc luminescence to less than a half of that of only PKR overexpression, again showing how IFI27 potentiates PKR activity (Fig 8B). On the other cell lines (PACT KO, PKR KO and DKO), the reduction observed after PKR and IFI27 overexpression compared to PKR overexpression alone, was very slight, and always lower than in the WT cells (Fig 8B). Interestingly, after poly(I:C) transfection, we only observed the reduction effect on translation after PKR and IFI27 overexpression, compared to PKR overexpression alone, on HeLa WT and HeLa PKR KO cells (Fig 8B). PKR overexpression was able to counteract PKR absence on PKR KO cells, thus allowing IFI27 to further activate PKR. However, when PACT is absent (PACT KO and DKO cells), the overexpression of PKR together with IFI27, has no effect compared to PKR overexpression alone (Fig 8B), showing the essential role of PACT on IFI27-mediated PKR activation, and showing that the expression of PKR plays a role in the translation inhibition mediated by IFI27.

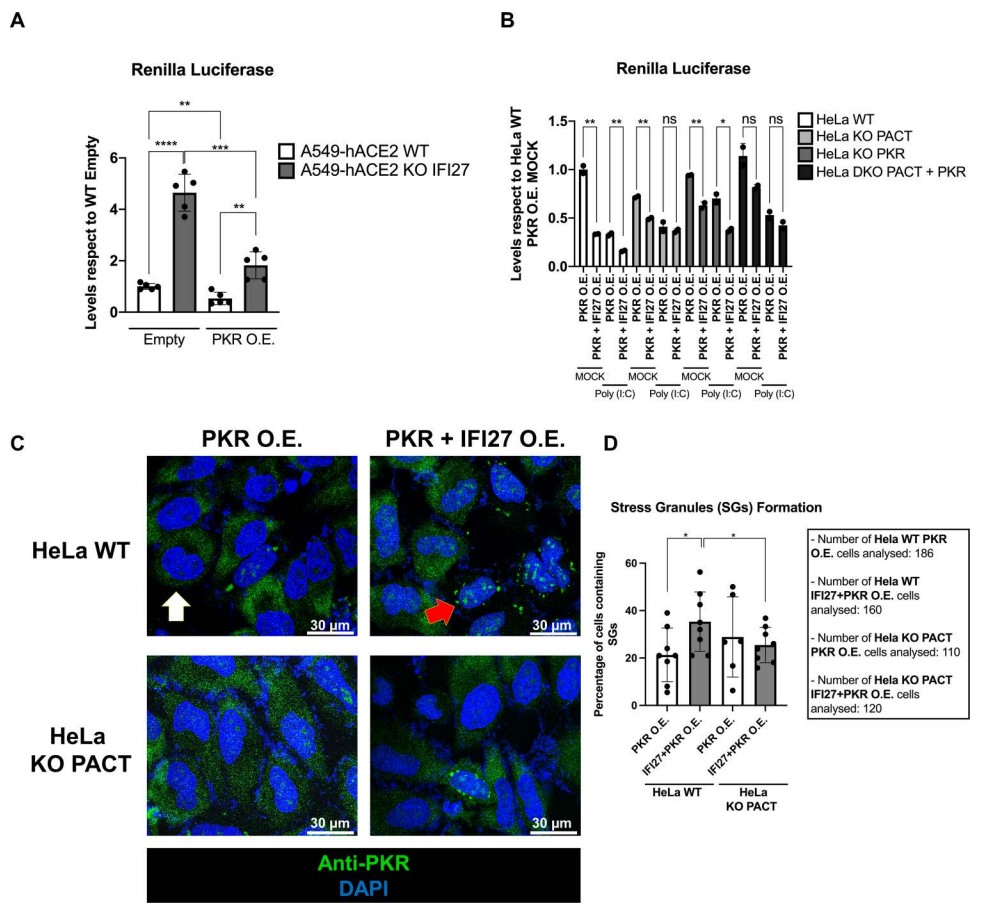

**Fig 8. IFI27 promotes the activation after PKR overexpression in a PACT-dependent manner. (A)** A549-hACE2 WT and A549-hACE2 IFI27 KO cells were transiently transfected with a pCAGGS-PKR-myc plasmid (PKR O.E.) or an Empty pCAGGS plasmid (Empty) in combination with an Rluc expressing pRL plasmid. 24h later, protein extracts were obtained by lysis and the levels of RLuc luminescence were measured. All luminescence levels were compared with respect to A549-hACE2 WT cells transfected with the empty plasmid. Individual data points correspond to independent measures from different technical samples. **(B)** HeLa WT, PACT KO, PKR KO or double KO for PACT and PKR (DKO) were transiently transfected with a pCAGGS-PKR-myc plasmid (PKR O.E.) alone or in combination with a pCAGGS-IFI27-HA plasmid (PKR + IFI27 O.E.). These plasmids were co-transfected together with a pRL plasmid expressing RLuc luciferase, and 24h later, the cells were left mock-transfected of transfected with poly(I:C) at 2 μg/ml for 7 hours. Then, protein extracts were obtained by lysis and the levels of RLuc luminescence were measured. All luminescence levels were compared respect to HeLa WT cells overexpressing PKR. Individual data points correspond to independent measures from different technical samples. Two independent experiments were performed with similar results. **(C)** HeLa WT and HeLa PACT KO cells were transiently transfected with either a pCAGGS-PKR-myc plasmid alone (PKR O.E.) or in combination with a pCAGGS-IFI27-HA (PKR + IFI27 O.E.) plasmid, and 24 hours later, transfected with poly(I:C) at 2 μg/ml for 7 hours. Then, cells were fixed with formaldehyde and permeabilized, and an immunofluorescence was performed for PKR (in green) and nuclei were stained with DAPI (in blue). White arrow **(C)** shows a cell with no presence of SGs, while the red arrow **(C)** shows a cell with positive presence of SGs. **(D)** Quantification of the percentage of SGs positive cells within the total number of counted cells for each condition is represented. Each dot indicates the percentage of SGs positive cells within a single region, having each region several cells. The total number of counted cells was 186 and 160 cells for WT cells transfected with the PKR plasmid alone, or with the PKR plasmid and the plasmid for IFI27 overexpression together, respectively; and 110 and 120 cells for PACT KO cells transfected with the PKR plasmid alone, or with the PKR plasmid and the plasmid for IFI27 overexpression together, respectively. Scale bar, 30 μm. Data is represented as the mean and standard deviations of the different measures. ns (non-significant) $p > 0.05$, *$p < 0.05$, **$p < 0.01$, ***$p < 0.001$, **** $p < 0.0001$ (for comparisons using unpaired two-tailed Student's t test in **A**, **B** and **D**).

Next, the formation of SGs was analyzed in cells, after overexpressing both PKR and IFI27, compared to cells overexpressing PKR alone. To this end, we transfected both HeLa WT and HeLa PACT KO cells with plasmids expressing PKR and IFI27 together, or with a plasmid expressing PKR alone, as stated before. At 24 hours after plasmid transfection, we treated these cells with poly(I:C) for 7 hours, and then, the cells were fixed and processed for immunofluorescence.

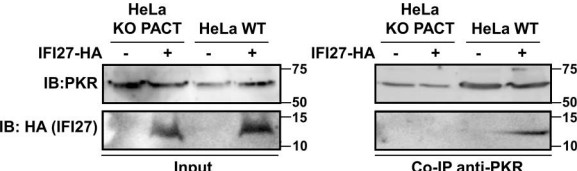

**Fig 9. IFI27 interacts with PKR in a PACT-dependent manner.** HeLa WT and HeLa PACT KO cells were transiently transfected with a pCAGGS-IFI27-HA plasmid, and 24h later, the cells were transfected with poly(I:C) at 3000 ng/ml. At 24h after poly(I:C) transfection, cell lysates were incubated overnight at 4°C with the PKR specific antibody together with protein A-sepharose resin. Eluates were analysed by Western blot, to detect PKR and IFI27 by employing anti-PKR (to detect endogenous PKR, top panel) and anti-HA (to detect IFI27-HA, bottom panel) both in the cellular lysates (Input) and after the Co-IP. Molecular weight is indicated on the right of the panels (in kilodaltons). These co-IPs were performed twice, in independent experiments.

Similar to Fig 7B and 7C, on HeLa WT cells, after overexpressing PKR together with IFI27, the percentage of SG-containing cells was of 34.4%, a percentage significantly higher than when PKR was overexpressed alone, which was of 21.5% (Fig 8C and 8D). This correlates with the reduction on RLuc luminescence on HeLa WT cells overexpressing PKR and IFI27 together, compared to cells overexpressing PKR alone (Fig 8B). Similar to Fig 7B, on HeLa PACT KO cells, the overexpression of PKR together with IFI27 did not induce a higher percentage of SG-containing cells, compared to over-expressing PKR alone (Fig 8C and 8D), showing again the importance of PACT on the positive effect of IFI27 on modulating PKR activation.

### IFI27 interacts with PKR in a PACT-dependent manner

Given the importance of PACT on the positive effect of IFI27 on modulating PKR activation (Figs 6–8), and the higher emPAI values obtained in the MS experiment for PACT than for PKR (Fig 1A), we hypothesized that the binding of IFI27 to PKR could be mediated by PACT. To test this hypothesis, HeLa WT and HeLa PACT KO cells were transfected with the pCAGGS-IFI27-HA plasmid, or with the empty plasmid, as control. A specific PKR antibody was bound to protein A-sepharose resin, and the cellular extracts overexpressing IFI27-HA were used to test the binding of IFI27-HA to endogenous PKR. By this method, we were able to confirm the interaction between IFI27 and PKR by a Western blot analysis, in the HeLa WT cells, but not in the HeLa PACT KO cells (Fig 9). Interestingly, these data further confirms that IFI27 co-immunoprecipitates with PKR, and indicates that the interaction of IFI27 with PKR is dependent on PACT, providing a molecular mechanism for the need of PACT in the activation of PKR mediated by IFI27.

### Discussion

In this work, we describe for the first time the ability of IFI27 protein to induce activation of PKR, specifically in a PACT-dependent manner. However, we acknowledge that the experiments have been performed in established cell lines and more experiments in primary cells or even in vivo are required to provide a more comprehensive understanding. When activated, PKR induces the phosphorylation of the translation initiation factor eIF2α at its Serine 52 residue, which will result in a severe reduction of *de novo* protein synthesis [16]. This mechanism is an important response strategy to different types of cellular stress, such as misfolded proteins accumulation, starvation, heat shock, iron deficiency, changes in intracellular calcium, or viral infections [16,88]. Phosphorylation of eIF2α inhibits eIF2B-mediated exchange of eIF2-GDP to eIF2-GTP, an essential step to form the ternary complex (eIF2-GTP-Met tRNA$_i$), essential for translation initiation [89]. The regulation of the activity of PKR is very complex and not completely known yet. PKR is ubiquitously and constitutive expressed in all vertebrates, but its expression is further induced by interferons in the context of pathogen infections [13]. Inactive forms of PKR exist in a weak monomer-dimer equilibrium that is reversible, and overexpression of PKR can induce its own activation, as we confirmed in this work (Fig 7), being the hallmark of this activation the phosphorylation of

its Threonin 446 residue [22]. The primary regulator of PKR activity are dsRNAs, which can be inhibitors, neutral or activators, and within the lasts, we find viral RNAs, which will bind to the dsRBMs of PKR and induce a conformational change that leads to activation of PKR [22,90]. However, this regulation is much more complicated than that, as also several protein partners can regulate PKR function, both by activating or inhibiting, as for instance, PKR-activating protein, PACT [90].

We describe for the first time that IFI27 binds to both PKR and PACT (Fig 1). Furthermore, we show that the binding of IFI27 to PACT is mediated by dsRNAs or RNAs containing duplex regions (Fig 1). Supporting this result is the fact that the co-immunoprecipitation of IFI27 with PACT is enhanced in the presence of poly(I:C), an analog of dsRNAs, and to the fact that the pre-treatment of the cellular extracts with RNAses inhibits the co-immunoprecipitation of IFI27 with PACT (Fig 1A). In addition, both IFI27 [43] and PACT [26] are able to bind dsRNAs. IFI27 is an ISG, whose expression is highly induced by several viral infections [43,91–95], and we have previously described that this protein is able to interact with dsRNAs and negatively regulate the activity of both RIG-I and MDA-5 proteins [43,44]. These interactions will point to the formation of regulatory clusters of proteins, in which IFI27 could play a very important role by modulating both RLRs [43,44] and promoting PKR function (Figs 2–4) in a PACT-dependent manner (Figs 5–8). In addition, we show that the binding of IFI27 to PKR is mediated by PACT (Fig 9), providing an explanation for the need of PACT to augment IFI27-mediated, PKR activation. The interaction with PACT to modulate PKR function has already been described for other ISG in the context of viral infections. As mentioned before, under HIV-1 infections, PACT interacts with ADAR1, and both proteins inhibit PKR activation [37,96,97].

In this study, we find that IFI27 interacts with PACT and PKR, and that after SARS-CoV-2 and VSV infections, or after poly(I:C) treatment, IFI27 further activates PKR function, strongly suggesting that the effect of IFI27 on PKR function is broad and happens after different stimuli. In addition, we show that IFI27 overexpression at high levels, promotes PKR phosphorylation even in the absence of infection (Fig 2B), and in the absence of infection in conditions of PKR overexpression (Fig 7); and that the knock-out (Fig 3B) of IFI27 can decrease PKR phosphorylation. This could be due to the activation of PKR by endogenous dsRNAs (reviewed in [13]), including mitochondrial RNAs, non-coding RNAs [57], and dsRNAs formed by inverted Alu repeats, specifically during mitosis [58].

Both RLRs and PKR are activated by the presence of viral dsRNAs or viral RNAs containing duplex regions. Furthermore, even though these both arms of the innate immunity, RLRs and PKR, may be activated by the same substrate, their functions within antiviral responses are very different [98]. RLRs when activated, induce the production of interferons and proinflammatory cytokines, while PKR induces a host shut-off that strongly impairs cellular and viral translation, to avoid viral protein synthesis, but also affecting general cellular translation and promoting apoptosis [99,100]. Apoptosis induction is also a function shared by IFI27, which has been described as a proapoptotic factor by destabilizing mitochondrial membrane potential [42,101]. Furthermore, IFI27 augments the apoptosis induced by Tumor necrosis factor-related apoptosis-inducing ligand (TRAIL) [102]. Therefore, this newly described ability of IFI27 to increase PKR activation, could also give an additional explanation to its proapoptotic behaviour.

Previously, we showed that IFI27 negatively modulates the innate immune responses induced after viral infections, by modulating RIG-I and MDA5 activation [43,44]. In addition, in this work we show how IFI27 positively affects PKR activation. Interestingly, PKR activation could modulate the innate immune responses after viral infections by blocking the translation of antiviral and inflammatory proteins. In this sense, the decreased PKR activation in cells knocked-down or knocked-out for IFI27 could be contributing to an increased translation of innate immune response proteins, providing, therefore, an additional molecular mechanism to the effect of IFI27 in negatively modulating innate immune responses, as we previously published [43,44]. In fact, it has been shown that Hepatitis C Virus blocks interferon effector function by inducing PKR phosphorylation, leading to decreased translation of ISGs [39,40], and that translation efficiency of type I IFN and ISGs was significantly downregulated by PKR activation after Zika virus infection [41]. In addition, when we over-express IFI27, PKR is activated to a higher extent than in the control cells, and SARS-CoV-2 titers are slightly increased

(S1A Fig), as we previously published [43,44]. This effect may be explained by an IFI27-mediated effect on activating PKR and subsequently decreasing the translation of antiviral proteins, and an effect of IFI27 on modulating RLR (RIG-I and MDA5) activation [43,44], rather than by an antiviral effect of PKR on blocking viral gene expression. Although the effect of IFI27 on counteracting IFN-responses might led to decreased expression of PKR, as PKR is an IFN-induced gene [14] and IFI27 negatively modulates the IFN responses [43,44], PKR is also constitutively expressed, and we do not detect significant differences in the levels of total PKR among cells overexpressing IFI27 (Figs 3C, 3F, and 6B), IFI27 knocked-out cells (Fig 4B and 4D), IFI27 knocked-down cells (Fig 5D) and their corresponding control cells. Thus, indicating that, at least in these conditions, IFI27 does not significantly impact the levels of PKR expression, whereas it promotes PKR activation.

An important hallmark or PKR activation is the production of SGs. These structures can be seen as regulator hubs of innate immune responses against viruses, as multiple components of these responses can be found on SGs: sensors such as RLRs, effectors such as OAS/RNAseL or ADAR1, regulators such as tripartite motif containing 25 (TRIM25) or PACT, apoptosis regulators, viral components and regulators of host gene expression or cytokine mRNAs [75,103]. It is interesting that IFI27 overexpression can increase SGs formation (Figs 7 and 8), but only in the presence of PACT, as in HeLa PACT KO cells, IFI27 overexpression did not significantly augment SG´s formation (Figs 7 and 8). Furthermore, we were able to confirm the presence of PKR within these granules (Fig 7A), as previously described [75], but we did not observe a clear co-localization of IFI27 with the protein G3BP1, which is a marker for SG (Fig 7D). This could point to an IFI27-mediated upstream activation of PKR, which will then induce the formation of SGs and its translocation to these structures of antiviral response regulation.

Many viruses have evolved to encode PKR antagonist proteins as a mechanism to interfere or bypass host antiviral responses. A few of the multiple examples of proteins codified within viral genomes to antagonize PKR function are [14]: Vaccinia virus K3L, E3L and K1L proteins [104–106]; Herpes simplex 1 ICP34.5 or Us11 proteins [107,108]; Influenza A and B viruses NS1 protein [109]; SARS-CoV-2 nucleocapsid protein [24]. Therefore, upon infection with viruses not encoding PKR antagonists, such as viruses lacking these proteins, IFI27 may lead to a stronger modulation of PKR activation. There are other viruses which have not developed specific PKR inhibiting proteins, but still show mechanisms to counteract PKR action, such as Zika Virus (ZIKV) which utilize small non-coding RNAs that interact with PKR [41], as well as it interferes with SGs assembly [110].

All in all, PKR activation is one of the main antiviral pathways in vertebrates, as this is shown by its functional preservation and the enormous number of PKR antagonism strategies that most viral families present. Thus, this new PKR activating mechanism through IFI27 and PACT, allows us to delve deeper into the regulation of the PKR function. Being able to better understand this regulation would allow us to modulate its functionality in the context of viral infections. In the case of viruses that are more sensible to PKR, this strategy of activation could be very effective to restrict their replication, and in the case of viruses that are able to counteract PKR function, it could present us with a new option to try to switch the balance to a greater activation of PKR and therefore a greater restriction of viral replication and production.

## Materials and methods

### Cells and viruses

Human embryonic kidney 293T (HEK-293T, ATCC CRL-11268) and human lung epithelial carcinoma A549 (ATCC CCL-185) cells, were kindly provided by Prof. Luis Enjuanes (National Center for Biotechnology-CSIC, Madrid, Spain). HeLa M WT, HeLa M PACT knock-out (KO), HeLa M PKR KO and HeLa M PACT and PKR double KO (DKO) cells were kindly provided by Thomas Michiels (Université Catholique de Louvain, Brussels, Belgium) [17]. All the cells were grown at 37°C in air enriched with 5% $CO_2$ using Dulbecco's modified Eagle's medium (DMEM, Gibco) supplemented with 10% fetal bovine serum (Gibco), and 50 mg/ml gentamicin (Gibco). The overexpression of human ACE2 (hACE2) on HEK-293T and A549 cells was achieved by cells transduction with a retrovirus expressing hACE2 and a blasticidin resistance gene (kindly

provided by Pablo Gastaminza, National Center for Biotechnology-CSIC, Madrid, Spain). IFI27 KO generation on A549-hACE2 cells was achieved by our group as previously described [43]. The A549-hACE2 IFI27 KO cells stably transfected with an empty plasmid, and IFI27 KO cells stably expressing IFI27-HA by means of a linearized-pCAGGS-IFI27-HA plasmid transfection, were previously generated and described by our group [44]. The HEK-293T-hACE2 and A549-hACE2 cells were grown in the same media containing 5 and 2.5 µg/ml of blasticidin (ThermoFisher Scientific), respectively.

In this work we used Vesicular Stomatitis Virus, Indiana strain, which encodes the green fluorescent protein, GFP (rVSV-GFP) [111], as well as SARS-CoV-2, isolated from a patient at the beginning of the pandemia, kindly provided by prof. Luis Enjuanes (National Center for Biotechnology-CSIC, Madrid, Spain) [112]. Both viruses were grown in Vero E6 cells. rVSV-GFP and SARS-CoV-2 were titrated by plaque assay (plaque forming units, PFU/ml) in confluent monolayers of Vero E6 cells seeded in 24-well plates, as previously described [113,114].

## Plasmids

Polymerase II expression pCAGGS plasmids encoding IFI27 (GenBank accession number NM_001130080.3) fused to a C-terminal HA epitope tag (pCAGGS-IFI27-HA), and PACT (Protein Activator of Interferon Induced Protein Kinase EIF2AK2) fused to a C-terminal FLAG epitope tag (pCAGGS-PACT-FLAG, GenBank accession number NM_003690.5) were generated as previously described [43]. The pCAGGS plasmids expressing the IFI27 S63L and V82A were generated by cloning overlapping PCRs comprising the IFI27 open reading frames into the plasmid pCAGGS. These overlapping PCRs were generated using as template the pCAGGS-IFI27-HA plasmid and primers encoding the sequences for amino acid changes (available upon request). Polymerase II expression pCAGGS plasmid encoding PKR (GenBank accession number NM_001135651.3) fused to an N-terminal myc epitope tag (pCAGGS-PKR-myc) were generated by RT-PCR using total RNA isolated from the human cell line A549, and later cloned using standard techniques (primers available upon request). The plasmid pCAGGS expressing GFP (pCAGGS-GFP) was previously described [115]. The pRL plasmid expressing Rluc under the early SV40 enhancer/promoter region (pRL-Rluc) was obtained from Promega.

## Co-immunoprecipitation assays

Human HEK-293T cells (100 mm-plate) were transiently transfected with a pCAGGS-IFI27-HA expressing plasmid, and 24 hours later, cells were treated with 3000 ng/ml of poly(I:C) using polyethylenimine (PEI, polysciences) for 16 hours. After this treatment, cells were lysed in the coimmunoprecipitation buffer (NaCl 250 mM; EDTA 1 mM; 50 mM TrisHCl, pH 7.5; NP-40 0.5%) containing protease (ThermoFisher Scientific) and phosphatase (Merck) inhibitors, sonicated for 5 minutes in an ultrasonic bath sonicator (Fisherbrand FB15051), and cleared by centrifugation. Cleared cell lysates were incubated overnight at 4ºC with the anti-HA affinity resin (Pierce, 26181) to retain IFI27-HA and IFI27-bound proteins. The cellular extracts combined with the affinity resins were washed three times in Tris-buffered saline (TBS, 25 mM Tris, 150 mM NaCl) tampon containing 0.1% of sodium dodecyl sulfate (SDS). To unbind proteins from the resin, 0.1 M glycine buffer at pH 2.4 was employed. IFI27-HA protein interactome was analysed by mass spectrometry and interpreted at the proteomics scientific service at National Centre for Biotechnology, as indicated below.

For further analysis of IFI27-HA and PACT-FLAG interaction, the HEK-293T cells (100 mm-plate) were transiently transfected with pCAGGS-IFI27-HA, pCAGGS-IFI27-S63L-HA or pCAGGS IFI27-V82A expressing plasmids, alone or together with a pCAGGS-PACT-FLAG or a pCAGGS empty plasmid, to maintain the amount of plasmid DNA costant, and 24 hours later, cells were left mock-treated or were treated with poly(I:C) at 3000 ng/ml for 24 hours. Alternatively, HEK-293T-hACE2 cells were transfected with pCAGGS-IFI27-HA, and pCAGGS-PACT-FLAG, and 24 hours later, the cells were infected with SARS-CoV-2 (MOI 1) during 24 h. After these treatments, cells were lysed in the coimmunoprecipitation buffer, sonicated and cleared by centrifugation. Where indicated, cellular lysates were treated with RNAseA (10 U/ml), RNAse T1 (400 U/ml) and RNAse III (10 U/ml), for 1 h at 37ºC, as previously reported [43,116]. Cleared cell lysates were incubated overnight at 4ºC with the anti-HA affinity resin (Pierce, 26181) to retain IFI27-HA, IFI27-HA variants and IFI27

bounded proteins, or, where indicated, with the anti-FLAG affinity resin (Merck, A2220), to retain PACT-FLAG. To unbind proteins from the resin, 0.1 M glycine buffer at pH 2.4 was employed. Eluted proteins were denatured in loading buffer and incubated at 95°C during 5 min. Then, an electrophoresis and Western blot were performed as described below.

Alternatively, to further analyze IFI27-HA and PKR interaction, HEK-293T cells, HeLa WT, and HeLa PACT KO cells (100 mm-plate) were transiently transfected with a pCAGGS-IFI27-HA expressing plasmid and 24 hours later, cells were treated with poly(I:C) at 3000 ng/ml for 16 hours. Then, the cells were lysed in the coimmunoprecipitation buffer, sonicated and cleared by centrifugation. Cleared cell lysates were incubated overnight at 4°C with the PKR specific antibody (sc-6282, Santa Cruz Biotechnology) as well as with protein A-sepharose (P3391, Merck) resin . Elution was performed with the 0.1 M glycine buffer at pH 2.4, and analysis was done by electrophoresis and Western blot as described below.

### Binding of IFI27 variants to poly(I:C)

Human HEK-293T cells (6-well plate format) were transiently transfected with the plasmids expressing the IFI27-HA variants, a plasmid expressing GFP (pCAGGS-GFP), or the plasmid expressing PACT (pCAGGS-PACT-FLAG), using lipofectamine 3000 (ThermoFisher Scientific). At 24 h, cells were lysed in the coimmunoprecipitation buffer, sonicated and cleared by centrifugation, and the cellular extracts were bound to poly(I:C)-conjugated agarose beads. To prepare these beads, 6 mg of poly(C)-conjugated agarose beads (Sigma) per sample were washed five times with TBS buffer, and the beads were then resuspended in buffer containing 50 mM Tris and 50 mM NaCl, and incubated with 120 µg of inosinic acid (Sigma) overnight at 4°C in a rotator. Next day, beads were washed twice with TBS, resuspended in TBS buffer containing 1 mM EDTA and 0.5% Triton X-100, and incubated at 4°C during 3 h with the cellular extracts expressing the IFI27 variants, GFP, or PACT, and with RNAse inhibitor (Promega, N251B). The mixture was then washed 4 times with TBS buffer containing 1 mM EDTA and 0.1% Tween 20, and the proteins bound to the matrix were eluted in loading buffer (Bio-Rad) containing 2.5% β-mercaptoethanol at 95°C during 5 min. The eluted proteins were analyzed by Western blot, as described below.

### Western blot assays

Cell lysates or immunoprecipitation eluates were mixed with Laemmli sample buffer (Biorad) containing 2.5% β-mercaptoethanol, and then heated for 5 minutes at 95°C to achieve protein denaturation. These samples were used for SDS-polyacrylamide gel electrophoresis (PAGE), under denaturing conditions. After SDS-PAGE, proteins were transferred to nitrocellulose membranes (Biorad). Membranes were blocked with either 5% nonfat dry milk or 5% bovine serum albumin (for phosphorylated protein detection) in TBS containing 0.1% Tween-20 during 1 hour at RT. Proteins were detected using a 1:1000 dilution of the primary antibodies in blocking buffer: anti-FLAG (F3165, Sigma-Aldrich) to detect FLAG-tagged PACT, anti-HA (H6908, Sigma-Aldrich) to detect HA-tagged IFI27 variants, anti-PACT (sc-377103, Santa Cruz Biotechnology), anti-eIF2α-P (9721, Cell Signaling Technology) to detect phosphorylated eIF2α, anti-eIF2α (9722, Cell Signaling Technology), anti PKR-P (ab32036, Abcam) to detect phosphorylated PKR, anti-PKR (sc-6282, Santa Cruz Biotechnology), anti-GFP (11814460001, Merck), anti-SARS-CoV-2 nucleoprotein (GTX135357, GeneTex), anti-GAPDH (sc-47724, Santa Cruz Biotechnology) and anti-β-actin (A4700, Sigma-Aldrich). After, membranes were incubated 1 hour at RT with goat anti-rabbit polyclonal antibodies (pAb) or goat anti-mouse monoclonal antibodies (mAb) conjugated to horseradish peroxidase (Sigma-Aldrich), diluted 1:4,000 on either 5% nonfat dry milk or 5% bovine serum albumin (for phosphorylated protein detection) in TBS containing 0.1% Tween-20. Membranes were revealed by chemiluminescence, using SuperSignal west femto maximum sensivity substrate (ThermoFisher Scientific).

### Mass spectrometry assays

Samples were first reduced with 50 mM TCEP (pH 8.0) for 60 min at 37°C and alkylated with 200 mM methyl methanethiosulphonate (MMTS, Pierce) for 10 min at room temperature. Subsequently, S-trap microcolumns (PROTIFI) were used to improve the tryptic digestion yield of low-abundance samples, as previously reported [117]. For protein identification

by tandem mass spectrometry (LC–MS/MS Exploris 240), the peptide samples were analyzed on a nano liquid chromatography system (Ultimate 3000 nano HPLC system, Thermo Fisher Scientific) coupled to an Orbitrap Exploris 240 mass spectrometer (Thermo Fisher Scientific). Samples (5 µL) were injected on a C18 PepMap trap column (5 µm, 100 µm I.D. x 2 cm, Thermo Scientific) at 30 µL/min, in 0.1% formic acid in water, and the trap column was switched on-line to a C18 PepMap Easy-spray analytical column (2 µm, 100 Å, 75 µm I.D. x 50 cm, Thermo Scientific). Equilibration was done in mobile phase A (0.1% formic acid in water), and peptide elution was achieved in a 30 min gradient from 4% - 35% B (0.1% formic acid in 80% acetonitrile) at 300 nL/min. Data acquisition was performed using a data-dependent top-25 method, in full scan positive mode (range of 350–1200 m/z). Survey scans were acquired at a resolution of 60,000 at m/z 200, with Normalized Automatic Gain Control (AGC) target of 300% and a maximum injection time (IT) = Auto. The top 25 most intense ions from each MS1 scan were selected and fragmented by Higher-energy collisional dissociation (HCD) of 30. Resolution for HCD spectra was set to 15,000 at m/z 200, with AGC target of 75% and maximum ion injection time = Auto. Precursor ions with single, unassigned, or six and higher charge states from fragmentation selection were excluded.

MS and MS/MS raw data were translated to mascot general file (mgf) format and searched using an in-house Mascot Server v. 2.7 (Matrix Science, London, U.K.) against a human database (reference proteome from Uniprot Knowledgebase). Search parameters considered fixed MMTS alkylation of cysteine, and the following variable modifications: methionine oxidation, pyroglutamic acid from glutamine and glutamic acid at the peptide N-terminus, deamidation of asparagine/glutamine and acetylation of the protein N-terminus. Peptide mass tolerance was set to 10 ppm and 0.02 Da, in MS and MS/MS mode, respectively, and 3 missed cleavages were allowed. The Mascot confidence interval for protein identification was set to ≥ 95% (P < 0.05) and only peptides with a significant individual ion score of at least 20 were considered. Furthermore, for our analysis purposes, only peptides with a significant individual ion score of at least 40 were considered (S1 Table).

### Immunofluorescence and confocal microscopy

HEK-293T, HeLa WT or HeLa PACT KO cells were seeded on sterile glass coverslips (24-well plates) at a confluency of 70%. Then, when indicated, cells were transiently transfected with the specific plasmid indicated for each experiment: pCAGGS-IFI27-HA at 600 ng per well (1 ng/µl) or pCAGGS-PACT-FLAG at 300 ng per well (0.5 ng/µl); and pCAGGS-EMPTY when necessary to transfect the same amount of DNA plasmid per well in each condition. When indicated, at 24 hours post-transfection (hpt) cells were left mock-treated or poly(I:C)-treated at the indicated concentration and duration (2 µg/ml for 7 hours for HeLa cells; 3000 ng/ml for 24 hours for HEK-293T cells). After mock or poly(I:C) treatments, cells were fixed and permeabilized with 10% formaldehyde and 0.1% Triton-X100 in PBS for 20 minutes at RT. Then, cells were incubated for 1 hour at RT in blocking buffer (10% fetal bovine serum, 0.1% Tween-20, 0.1% Triton-X100 in PBS). After blocking, the cells were incubated with the specific primary antibodies overnight at 4°C. The primary antibodies were diluted in phosphate-buffered saline (PBS) containing 0.1% Tween-20 as following: anti-HA antibody generated in rabbit (H6908, Sigma-Aldrich, at 1:1000), anti-FLAG antibody generated in mouse (F3165, Sigma-Aldrich, at 1:1000), anti-PKR antibody generated in mouse (sc-6282, Santa Cruz Biotechnology, at 1:500), anti-PKR antibody generated in rabbit (18244–1-AP, Protein Tech, at 1:500), anti-eIF3η antibody generated in mouse (sc-137214, Santa Cruz Biotechnology, at 1:500), and anti-G3BP1 antibody generated in mouse (sc-365338, Santa Cruz Biotechnology, at 1:500). After incubation, coverslips were washed 4 times with PBS and stained with the secondary anti-mouse or anti-rabbit antibodies, conjugated to Alexa Fluor 488 or Alexa Fluor 594 (Invitrogen) respectively, for 45 minutes at RT. After, coverslips were washed 4 times with PBS, and then incubated with DAPI (ThermoFisher Scientific) for 15 minutes to achieve nuclei staining. Coverslips were mounted on microscope slides using ProLong Gold antifade reagent (Invitrogen) and analyzed on a Leica STELLARIS 5 confocal microscopy. The same instrument settings were applied to all images, and the images were analyzed using Fiji software.

## Translation shut-down assays

To evaluate the level of translation activity within cells, HEK-293T-hACE2, HeLa (WT, PACT KO, PKR KO or DKO), A549-hACE2 (WT or IFI27 KO) cells, A549 IFI27 KO stably transfected with an empty plasmid or A549 IFI27 KO stably expressing IFI27-HA by means of a linearized plasmid transfection, were transiently transfected with pRL-Renilla luciferase (Rluc) in combination with other plasmids or silencing RNAs (siRNAs) as stated in each specific case. After transfections, cells were either mock-treated or poly(I:C)-treated. Alternatively, the cells were left mock-infected, or the cells were infected with SARS-CoV-2 or rVSV-GFP infected (concentrations, MOI and times as indicated in each specific case). After treatments or infections, protein lysates were obtained and the levels of RLuc luminescence were measured by Renilla Luciferase Assay System (Promega) according to manufacturer´s instructions and using a luminometer (Tecan infinite M200 pro). The mean values and standard deviations were calculated using GraphPad Prism, GraphPad Software, Boston, Massachusetts USA (www.graphpad.com).

## Silencing of IFI27

Human A549-hACE2 or HeLa WT cells were transfected with a small interfering RNA (siRNA) specific for IFI27 (ThermoFisher Scientific, s7140), or a negative control non-targeting siRNA (ThermoFisher Scientific, AM4635). This transfection was performed twice, 24 hours apart. All siRNAs were transfected at a final concentration of 20 nM and using lipofectamine RNAiMax (ThermoFisher Scientific) according to manufacturer´s instructions, as previously described [43].

## Quantitative PCR assay

mRNA levels of IFI27 were analysed in order to confirm that the IFI27 knockdown was successful. RLuc mRNA levels were analyzed to compare mRNA expression at the transcriptional level. For both ends, total RNA was extracted using total RNA extraction kit (Omega Biotek). Retrotranscriptase (RT) reactions were performed by employing High-Capacity cDNA transcription kit (ThermoFisher Scientific) at 37ºC for 2 hours, using random primers and total RNA as template. qPCRs are performed with cDNAs obtained from RT reactions. For IFI27 expression, TaqMan gene expression assays (Applied Biosystems) specific for human IFI27 (Hs01086373_g1) and human GAPDH (Hs02786624_g1) were used. For Rluc expression, qPCR was then performed using the SYBR Green PCR Master mix (Thermo Fisher Scientific) and the primers Rluc-F (5´- GCAGAAGTTGGTCGTGAGG-3´) and Rluc-R (5´-TCATCCGTTTCCTTTGTTCTG-3´), specific for Rluc open reading frame, as previously described [118], or the primers hGAPDH-F (5´-TGCACCACCAACTGCTTAGC-3´) and hGAPDH-R (5´-GGCATGGACTGTGGTCATGAG-3’), specific for GAPDH. To quantify, we used the thresholds cycle ($2-\Delta\Delta CT$) method [119] and normalized with GAPDH expression levels.

## Supporting information

**S1 Fig. PACT CoIP in SARS-CoV-2 infected cells.** (A and B) HEK-293T-hACE2 cells were transiently transfected either with a pCAGGS-PACT-FLAG plasmid alone, a pCAGGS IFI27-HA plasmid alone, both pCAGGS-PACT-FLAG and pCAGGS-IFI27-HA plasmids, or an emtpy pCAGGS plasmid (Empty) and 24h later, transfected cells were infected with SARS-CoV-2 at a multiplicity of infection (MOI) of 0.5 during 24 hours. At 24h after infection, protein extracts were obtained by lysis. (B) Protein extracts were analysed by Western blot (input) with antibodies specific for FLAG (to detect PACT-FLAG), HA (to detect IFI27-HA), SARS-CoV-2 nucleocapsid protein, and GAPDH. Molecular weights are indicated on the right of the panels (in kilodaltons). (B) Protein extracts were incubated with FLAG-bound agarose beads to retain PACT-FLAG and all its associated proteins. Eluates were analysed by Western blot, to detect PACT-FLAG and IFI27-HA by using anti-FLAG (to detect PACT-FLAG, top panel) and anti-HA (to detect IFI27-HA, bottom panel) antibodies after the Co-IP. Molecular weights are indicated on the right of the panels (in kilodaltons).
(TIF)

**S2 Fig. Viral titrations.** (A, B, C) HEK-293T cells were transiently transfected with a pCAGGS-IFI27-HA plasmid (IFI27 O.E.) or an emtpy pCAGGS plasmid (Empty) and 24h later, transfected cells were left mock-infected or infected with SARS-CoV-2 at a multiplicity of infection (MOI) of 1 during 24 hours (A), or infected with VSV at a MOI of 1 during 8 (B) or 24 hours (C). SARS-CoV-2 and VSV viral titers were determined by plaque assay (plaque forming units, PFU/ml) in confluent monolayers of Vero E6 cells seeded in 24-well plates, as previously described [43,44]. Data is represented as the mean and standard deviations of triplicate measures. ns (non-significant) $p > 0.05$, *$p < 0.05$, **$p < 0.01$, ***$p < 0.001$, **** $p < 0.0001$ (for comparisons using unpaired two-tailed Student's t test in A, B and C).
(TIF)

**S3 Fig. Dose-dependent effect of IFI27-HA expression on PKR activation.** (A, B, C, D) HEK-293T cells were transiently transfected with a pCAGGS-IFI27-HA plasmid (IFI27 O.E.) or an emtpy pCAGGS plasmid (Empty) together with a pRL plasmid expressing RLuc luciferase, and 24h later, transfected cells were left mock-infected (A, B, D) or infected with VSV (A, C, D) at a multiplicity of infection (MOI) of 0.1 during 24 hours. 24 hpi, protein extracts were obtained, and RLuc luminiscence was measured (A). These protein extracts were also used to measure protein levels of IFI27-HA and β-actin by Western blot employing their respective antibodies. Molecular weight is indicated on the right of the panels (in kilodaltons). Western blots were quantified by densitometry using ImageJ software. The amount of IFI27-HA was normalised to the amounts of β-actin (results of quantification showed in bars below the immunoblots, with levels relative to MOCK 0.25 ng/μl). Data is represented as the means and standard deviations of triplicate measures. $p > 0.05$, *$p < 0.05$, **$p < 0.01$, ***$p < 0.001$, **** $p < 0.0001$ (for comparisons using unpaired two-tailed Student's t test in A). The asterisks above each bar represent the comparison vs. the control condition.
(TIF)

**S4 Fig. IFI27 knock-down.** A549-hACE2 WT cells were transfected twice, with a 24-hour gap between each transfection, either with a negative control non-targeting siRNA (siRNA Neg) or with an IFI27 siRNA (siRNA IFI27). 24 hours after the second transfection, cells were either mock or SARS-CoV-2 infected at a MOI of 1 for 24 hours. Total RNA was extracted, and a qRT-PCR was performed, to determine the level of expression of the IFI27 mRNA in each condition, comparing siRNA Neg and siRNA IFI27 effect on IFI27 expression. Data is represented as the mean and standard deviations of triplicate measures. ns (non-significant), $p > 0.05$, *$p < 0.05$, **$p < 0.01$, ***$p < 0.001$, **** $p < 0.0001$ (for comparisons using unpaired two-tailed Student's t test).
(TIF)

**S5 Fig. PKR protein expression.** HeLa WT and HeLa PKR KO cells were transiently transfected with a pCAGGS-IFI27-HA plasmid (IFI27 O.E.) or an empty pCAGGS plasmid (empty). 24 hours post-transfection, protein extracts were obtained by lysis and the protein levels of total PKR and β-actin were measured by Western blot employing their respective antibodies. Molecular weight is indicated on the right of the panels (in kilodaltons).
(TIF)

**S6 Fig. Renilla luciferase expression in WT and IFI27 KO cells.** A549-hACE2 WT and A549-hACE2 IFI27 KO cells were transiently transfected with a pCAGGS-PKR-myc plasmid (PKR O.E.) or an Empty pCAGGS plasmid (Empty) in combination with an Rluc expressing pRL plasmid. At 24h later, the cellular RNAs were purified and the levels of RLuc mRNAs were measured by RT-qPCR. Data are represented as the mean and standard deviations of triplicate measures. ns (non-significant).
(TIF)

**S1 Table. IFI27-HA-bound proteins identified by mass spectrometry.** HEK-293T cells were transfected with a pCAGGS plasmid encoding for IFI27 fused to HA tag (pCAGGS-IFI27-HA) or with a pCAGGS empty plasmid, as control.

24 hours later, cells were treated with Poly(I:C) at 3000 ng/ml. At 24h after poly(I:C) transfection, protein extracts were obtained by lysis and were incubated with HA-bound agarose beads to retain IFI27-HA and all its associated proteins, which were then identified by MS. The proteins identified only in the IFI27-HA-overexpressing cells are shown. Protein accession number (prot_acc), protein description (prot_desc), prot_score, prot_mass, prot_matches_sig, prot_sequences_sig, prot_coverage and prot_PI (isoelectric point) are indicated, being these terms previously explained in (https://www.matrixscience.com/help/csv_headers.html).
(DOCX)

**S2 Table. Biological process organization of mass spectrometry data.** All proteins found on IFI27 interactome with a score higher than 40 (S1 Table) were analysed by Gene Ontology, using NIH DAVID software (https://davidbioinformatics.nih.gov/). Proteins were classified by biological process, and we selected all those groups of proteins whose function showed a False Discovery Rate (FDR) lower than 0.05. Category of classification, term of biological process, count of proteins within that biological process, percentage of the total of proteins represented by that biological process, pvalue, Fold Enrichment and False Discovery Rate (FDR) are indicated.
(DOCX)

**S3 Table. Molecular functions organization of mass spectrometry data.** All proteins found on IFI27 interactome with a score higher than 40 (S1 Table) were analysed by Gene Ontology, using NIH DAVID software (https://davidbioinformatics.nih.gov/). Proteins were classified by molecular functions, and we selected all those groups of proteins whose function showed a False Discovery Rate (FDR) lower than 0.05. Category of classification, term of molecular function, count of proteins within that molecular function, percentage of the total of proteins represented by that molecular function, pvalue, Fold Enrichment and False Discovery Rate (FDR) are indicated.
(DOCX)

## Acknowledgments

The proteomic analysis was performed in the proteomics facility of the National Center for Biotechnology-CSIC, Madrid, Spain. We thank Prof. Thomas Michiels (Université Catolique de Louvain, Brussels, Belgium), for kindly providing us the Hela M WT, PACT KO, PKR KO and DKO cells.

## Author contributions

**Conceptualization:** Marta DeDiego.

**Data curation:** Darío López-García.

**Formal analysis:** Darío López-García, Marta DeDiego.

**Funding acquisition:** Marta DeDiego.

**Investigation:** Darío López-García, Vanessa Rivero, Laura Villamayor, Marta DeDiego.

**Methodology:** Darío López-García, Vanessa Rivero, Laura Villamayor, Marta DeDiego.

**Project administration:** Marta DeDiego.

**Resources:** Marta DeDiego.

**Supervision:** Marta DeDiego.

**Validation:** Darío López-García, Vanessa Rivero, Laura Villamayor, Marta DeDiego.

**Writing – original draft:** Darío López-García, Marta DeDiego.

**Writing – review & editing:** Darío López-García, Vanessa Rivero, Laura Villamayor, Marta DeDiego.

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
