## [Decision Letter · Decision Letter 0]

PPATHOGENS-D-24-02606

IFN alpha inducible protein 27 (IFI27) acts as a positive regulator of PACT-dependent PKR activation

PLOS Pathogens

Dear Dr. L. DeDiego,

Thank you for submitting your manuscript to PLOS Pathogens. After careful consideration, we feel that it has merit but does not fully meet PLOS Pathogens's publication criteria as it currently stands. Therefore, we invite you to submit a revised version of the manuscript that carefully addresses the points raised during the review process.

Please submit your revised manuscript within 60 days Mar 01 2025 11:59PM. If you will need more time than this to complete your revisions, please reply to this message or contact the journal office at plospathogens@plos.org. Please include the following items when submitting your revised manuscript:

We look forward to receiving your revised manuscript.

Kind regards,

Peter Sarnow

Academic Editor

PLOS Pathogens

Alexander Gorbalenya

Section Editor

PLOS Pathogens

 Sumita Bhaduri-McIntosh

Editor-in-Chief

PLOS Pathogens

orcid.org/0000-0003-2946-9497 

Michael Malim

Editor-in-Chief

PLOS Pathogens

orcid.org/0000-0002-7699-2064

**Additional Editor Comments:**

Please consider acknowledging RNA virus infection(s) in the title.

**Journal Requirements:**

1) We notice that your supplementary Figures are included in the manuscript file. Please remove them and upload them with the file type 'Supporting Information'. Please ensure that each Supporting Information file has a legend listed in the manuscript after the references list.

2) Please ensure that the funders and grant numbers match between the Financial Disclosure field and the Funding Information tab in your submission form. Note that the funders must be provided in the same order in both places as well. State what role the funders took in the study. If the funders had no role in your study, please state: "The funders had no role in study design, data collection and analysis, decision to publish, or preparation of the manuscript.".

**Reviewers' Comments:**

Reviewer's Responses to Questions

**Part I - Summary**

Reviewer #1: The authors have investigated the role of IFI27 in PKR mediated stress responses including translational control and stress granule formation. They have shown that IFI27 interacts with PKR and PACT using high throughput mass spectrometry and co-immunoprecipitation experiments. They demonstrate that overexpression of IFI27 can promote PKR activity and inhibit cellular translation and stress granule formation while depletion of IFI27 inhibits PKR activity and stimulates cellular translation and stress granule formation. They further show that PACT is important in mediating these responses using double knockouts. However, the effects are marginal in several places and the quantification plots do not reflect what is seen in the western blot/ microscopy images. Importantly, it is unclear why IFI27 is able to modulate translation even in the absence of poly I:C stimulation or PKR activation given that IFI27 can only interact with PACT in a RNA-dependent manner.

Reviewer #2: The manuscript by Lopez-Garcia et al. describes the identification of IFI27 as a putative novel interacting protein for dsRNA-dependent PACT-mediated PKR activation.

The primary identification of IFI27-specific protein-protein interactions (PPI) were based on overexpressed HA-tagged IFI27 in HEK 293T cells followed by mass spectrometry. Subsequently, the authors performed multiple gain- and loss-of protein function experiments in presence and absence of viral or synthetic RNA molecules to provide evidence for IFI27 functions.

The manuscript is overall well written with some minor language issues. Although the identification of IFI27 is interesting and relevant, I find the manuscript quite preliminary. The impact of the finding is limited due to the application of transfection methods in all experiments. In addition, the authors did not consider dose- and time-dependent effects which I find crucial when investigating an apoptosis-inducing signaling pathway. In would also recommend to perform a mutagenesis study to define the IFI27 interacting domains to provide impactful and a complete data set.

Reviewer #3: The study titled "IFN alpha inducible protein 27 (IFI27) acts as a positive regulator of PACT-dependent PKR activation" explores a novel mechanism in the regulation of the antiviral protein kinase R (PKR), which is activated during viral infections. PKR plays a critical role in the innate immune response by inhibiting protein translation upon activation, impairing viral replication, and modulating host immunity.

This manuscript offers significant contributions to the understanding of PKR regulation in antiviral defense. The work presents a compelling and innovative exploration of the role of IFI27 as a positive regulator of PACT-dependent PKR activation, significantly advancing our understanding of antiviral immunity.

This work not only highlights a novel mechanism of PKR activation but also opens avenues for potential therapeutic interventions targeting RNA-virus infections. The clarity and depth of the research make it a valuable contribution to the field of innate immunity and virology.

**Part II – Major Issues: Key Experiments Required for Acceptance**

Reviewer #1: Introduction:

Shorten introduction to clearly define hypothesis- The rationale behind the study is not clear- Do the authors want to study IFI27 functions in general or only with respect to its ability to modulate PKR and PACT? Why do you hypothesize that IFI27 will modulate PKR/PACT functions in the first place?

Fig 1A: Complete mass spec data should be made available as a supplementary table so that the ranking of PKR and PACT can be seen. The authors don’t comment on any of the other proteins involved in PKR/PACT mediated pathwyas or translation control pathways that might have appeared in the mass spec results and could be regulated by IFI27.

Fig1B: Why was endogenous IFI27 not pulled down after poly I:C stimulation?

Fig 1C: Is the interaction between IFI27 and PKR RNA-dependent?

Fig 2A: Add a schematic of the luciferase constructs and the experimental pipeline above the graph for better understanding.

Fig 2B:

a. Why are mock and SARS-infected lysates run on separate gels? This doesn’t allow us to compare levels of P-eIF2-alpha, P-PKR and PACT between mock and infected cells. Since the blots are separate, band intensity quantification can’t be combined into one.

b. What happens to endogenous IFI27 levels upon infection?

c. How is IFI27 overexpression able to increase levels of P-PKR and P-eIF2-alpha even in the absence of dsRNA in mock cells, given that its interactions with PACT is RNA dependent?

Fig2D:

a. Why is IFI27 expression so much higher in VSV infected cells at 8hpi compared to 24hpi?

b. Mock- infected cells should be included in the blot.

c. The changes in the blot appear to be very marginal and densitometry doesn’t reflect this. Error bars in the densitometry graphs are also not clear. Western blots and densitometries need to be repeated.

Fig 3B: Does expression of IFI27 in KO cells rescue phenotype?

Fig 3-4:

a. Western for IFI27 is missing in KO/KD experiments to test if the protein is depleted.

b. Why is luciferase activity increasing in mock cells upon IFI27 KD when there is no change in P- PKR or P-eIF2-alpha levels. This is also contradictory to what was seen in IFI27 overexpressing mock cells in Fig 2B.

c. Figures 3 and 4 can be combined into one.

Fig 4B: Can PACT overexpression rescue the phenotype?

Fig 5: Why is luciferase decreasing in mock cells upon IFI27 overexpression when there is no change in P-PKR or P-eIF2-alpha?

Fig 6: Once again, the microcopy images are not reflective of what is seen with quantification. Increase in G3BP1 or eIF3A is not evident in the images shown.

Fig 6 and 7 convey the same thing and can be combined into one.

Fig 8: Images need be better representative

Reviewer #2: 1. The initial experiment was done by transfecting HA-tagged IFI27 in HEK-293T cells which have low level MDA5. Since MDA5 can activate PACT, the very first experiment already includes an experimental bias. In addition, the authors should rule out any effect of the HA-tag by confirming the results with a different tagging and untagged versions. Please also confirm the finding in at least a second cell line, preferably primary human cells (see point 2).

2. Transfections stress cells� Due to this exogenous stress, PACT might already be phosphorylated at positions S246 and S287, which increases PKR activation (see e.g. review PMID: 33387379. Please rule out that the transfection procedures do not include a bias. I would recommend to confirm endogenous PPI analyses. I would use +/- IFN treated cells (optimally primary cells, not cancer cells). IP could be done using commercial PKR antibodies for pull down and perform MS afterwards. This would show PPI under more native conditions avoiding transfection procedures.

3. Most presented data is based on single plasmid/virus/RNA concentrations and single times points. Since PACT/PKR activation causes translational shut off, the actual transfection procedures, the plasmid concentrations, and late time points might cause apoptosis. Please include time and dose-dependent experiments for some of the crucial findings. You might also perform cell viability assays to rule out apoptosis.

4. As model viruses, the authors use SARS-CoV-2 and VSV both triggering RIG-I and MDA-5 to certain levels. I would be recommended to use viruses (e.g EMCV for MDA-5) or exogenous stimuli that exclusively trigger MDA-5 or RIG-I to evaluate the effects of RIG-I and MDA5-dependent PACT activations. This is especially important as HEK293T have low level MDA5 expression (see comment above). I would also recommend to perform Westernblots for RIG-I and MDA5 to monitor proteins that influence downstream PACT activation.

5. Figure 2/Page 13: as the authors mention, SARS-CoV-2-mediated PKR activation has been well defined already. For confirmation and specification of the findings, I would strongly recommend to use a virus pair harboring defined mutations influencing PKR interactions e.g. SARS-CoV-2 N or nsp15 mutants or FluA NS1 mutants etc.. In addition, at least 2 time points would be necessary to observe the dynamics of the virus-mediated interferences. This is especially important as the authors use transfection-based assays that strongly influence the interferon and PKR-dependent pathways.

6. For an impactful publications, I would expect a mutagenesis study showing how exactly IFI27 interacts (at least domain level).

7. Figure 8. How do the authors explain the difference in RLUC expression between WT and IFI27 KO in mock treated cells? Did the authors confirm that transfection efficiency is equal between WT and KO? To confirm the specificity of the knockout, I would also suggest re-complementing IFI27 for example by transient overexpression in the KO cells.

8. Figure 2D. The signal to noise ratio in 2D seems unacceptable for actin. A normalization to this is not possible. The WB should be repeated if possible. I would probably use a different reference gene/protein as Actin expression/translation is often affected by virus infections see e.g. PMID: 15705200.

9. Figure 5A and 8B: Why does the KO of PACT (“empty”) lead to decreased protein translation. Shouldn't it have a similar effect as PKR KO, thereby increasing protein translation?

10. Figure 9. The authors show a dsRNA dependent interaction between IFI27 and PACT by including a RNAse condition in their co-IP (Figure 1). For the IFI27-PKR interaction in Figure 9 they omit the RNAse treatment and seem to claim a direct interaction. I would suggest that the authors include the RNAse treatment also in the IFI27-PKR co-IP to exclude dsRNA mediated interaction.

Reviewer #3: The manuscript could benefit from a more detailed discussion or additional experiments and controls:

- Mechanistic Details of IFI27 and PACT Interaction. While the study establishes that IFI27 interacts with PACT to activate PKR, the precise molecular mechanisms remain unclear.

1. In the immnoprecipitation experiments, the authors could analyze the interaction in the context of viral infection, not only using polyIC as a trigger.

2. In figure 1D, the authors should explicitly state whether the colocalization shown in Figure 1D occurs under mock-treated or poly(I:C)-treated conditions.

Perform colocalization analysis both before and after poly(I:C) treatment. This would provide valuable insights into how the interaction between IFI27, PACT, and PKR changes in response to dsRNA stimulation.

3. Immunofluorescence and colocalization images (fig 6 and 7 While stress granule (SG) formation is noted as a significant finding, the lack of clear co-localization of IFI27 with SG markers (e.g., G3BP1) raises questions about the direct role of IFI27 in SG formation. This requires further clarification and higher imaging quality (i.e. zoom areas or higher magnification).

4. The experiments focus primarily on HEK-293T, A549, and HeLa cell lines, which, while informative, may not fully capture the complexity of IFI27’s role in diverse cellular or tissue-specific environments. Expanding to primary cells or animal models would provide a more comprehensive understanding.

**Part III – Minor Issues: Editorial and Data Presentation Modifications**

Reviewer #1: 1. Quality of images and blots need to be better

2. Writing should be more concise and articulate especially where the authors describe quantification of stress granules in IFI27 expressing PACT KO cells

Reviewer #2: 1. Please show individual data points in all Figures. How often were experiments repeated independently?

2. Figure 6C is way too small. Figure 6 and figure 7 show the same thing with a slightly different analysis. Consider merging them. Figure 6B could be in the supplement

3. Figure 7. How many cells were counted and how often was the experiment repeated? Please show single data points to get estimates on the distributions.

4. Figure 8: Please explain the purpose and value of figure 8? As far as I understand it shows similar results that previous figures but just in cells that overexpress PKR. IFI27 increases the number of SGs dependent on PACT (same as fig 6) and a IFI27 KO increases protein translation (same as fig 3). In cells with artificially high PKR levels the PKR induction (and downstream translation inhibition) is slightly higher but, in my opinion, this does not add anything and is expected. It might be considered for the supplement.

Reviewer #3: 1. While the figures are informative, some lack detailed annotations or explanations, making interpretation more challenging for readers unfamiliar with the methods.

2. Statistical significance is noted but could benefit from expanded analysis, such as additional replicates, or indicating if they were done.

3. Potential Conflict with Previous Findings:The manuscript claims that IFI27 enhances the activity of PKR, promoting antiviral responses by reducing protein translation. This contrasts with the previous findings (Villamayor et al.,2023), where IFI27 was shown to negatively regulate innate immunity, particularly by interacting with Retinoic acid-inducible gene I (RIG-I) and melanoma differentiation-associated protein 5 (MDA-5). These interactions reportedly suppress downstream immune signaling, reducing interferon (IFN) production. Reconciling these roles with additional discussion,possible explanations, or experiments would enhance the paper's coherence.

4. While the authors tested IFI27's role using SARS-CoV-2 and Vesicular Stomatitis Virus (VSV), explaining their rationale for selecting these specific viruses over others would add context and strengthen their conclusions. In addition, what is the rational behing using one virus or the other in some of the experiments?.

5. In figure 2, the authors use only one time point (24 hpi) for SARS-CoV2 infections, while for VSV, the authors use two time points (8 and 24 hpi). Please explain.

PLOS authors have the option to publish the peer review history of their article (what does this mean? ). If published, this will include your full peer review and any attached files.

**Do you want your identity to be public for this peer review?** For information about this choice, including consent withdrawal, please see our Privacy Policy .

Reviewer #1: No

Reviewer #2: No

Reviewer #3: No

**Figure resubmission:**
---

## [Decision Letter · Decision Letter 1]

PPATHOGENS-D-24-02606R1

IFN alpha inducible protein 27 (IFI27) acts as a positive regulator of PACT-dependent PKR activation after RNA virus infections

PLOS Pathogens

Dear Dr. DeDiego,

Thank you for submitting your manuscript to PLOS Pathogens. After careful consideration, we feel that it has merit but does not fully meet PLOS Pathogens's publication criteria as it currently stands. Therefore, we invite you to submit a revised version of the manuscript that addresses the points raised during the revised review process.

Please submit your revised manuscript within 30 days Jul 04 2025 11:59PM. If you will need more time than this to complete your revisions, please reply to this message or contact the journal office at plospathogens@plos.org. Please include the following items when submitting your revised manuscript:

We look forward to receiving your revised manuscript.

Kind regards,

Peter Sarnow

Academic Editor

PLOS Pathogens

Alexander Gorbalenya

Section Editor

PLOS Pathogens

Sumita Bhaduri-McIntosh

Editor-in-Chief

PLOS Pathogens

orcid.org/0000-0003-2946-9497

Michael Malim

Editor-in-Chief

PLOS Pathogens

orcid.org/0000-0002-7699-2064

**Reviewers' Comments:**

Reviewer's Responses to Questions

**Part I - Summary**

Reviewer #1: The authors have successfully addressed all the concerns that I raised last time.

Reviewer #2: The authors addressed the concerns adequately. I have no further comments.

Reviewer #3: This manuscript explores a novel role for the interferon-stimulated gene IFI27 as a positive regulator of PKR activation through a PACT-dependent mechanism. The authors demonstrate that IFI27 interacts with both PACT and PKR in an RNA-dependent fashion, enhancing PKR phosphorylation and subsequent eIF2α phosphorylation, thereby suppressing protein translation. These effects are observed under basal conditions and during infection with RNA viruses, including SARS-CoV-2 and VSV, and are supported by a combination of proteomic analysis, co-immunoprecipitation, reporter assays, stress granule quantification, and genetic KO/KD models.

The authors have responded thoroughly to previous reviewer critiques. They provide a detailed dissection of the IFI27-PACT-PKR axis, including time- and dose-response analyses, confirmation of effects across multiple cell lines, and evidence for specificity using IFI27 mutants and rescue experiments. The revised version includes improved figure quality, better quantification, and additional experimental controls that address prior concerns about transfection bias and specificity.

The novelty lies in identifying IFI27 as a regulatory component of the PKR signaling pathway. Given the central role of PKR in translational control and innate immunity, these findings are both mechanistically important and of broad relevance to the pathogen-host interaction field. The work also opens potential avenues for therapeutic modulation of PKR activity during viral infections.

One limitation is the exclusive use of tumor-derived cell lines. While the authors acknowledge this and discuss it in the manuscript, future studies in primary cells or in vivo would be needed to fully generalize the findings. Another minor caveat is that while IFI27-PKR interactions are shown to be PACT-dependent, the RNA dependence of the IFI27-PKR complex is not directly tested with RNase treatment. Nonetheless, the presented evidence is convincing and well controlled.

Overall, the manuscript is clearly written, data-rich, and scientifically sound. It provides novel mechanistic insight into a relevant antiviral pathway and significantly improves upon the previous version.

**Part II – Major Issues: Key Experiments Required for Acceptance**

Reviewer #1: The western blots in Fig 1D seems to be inappropriately cropped and distinct bands are not seen in some of them, bands are merged into a single line. The blots should be replaced before final submission.

Reviewer #2: I have no further comments

Reviewer #3: The study already presents a robust and novel mechanistic model, and the current dataset supports the central conclusions:

1.Clarify the RNA dependence of IFI27–PKR interaction:

The authors demonstrate that the IFI27–PACT interaction is RNA-dependent (Fig. 1B), and that IFI27–PKR binding is PACT-dependent (Fig. 9), with enhanced association upon poly(I:C) stimulation (Fig. 1E). However, RNA dependence of the IFI27–PKR complex has not been directly tested via RNase treatment. Although the authors reasonably argue that the interaction is indirect via PACT, including RNase treatment in the PKR co-IP would definitively support this model. If technically feasible, this addition would complete the mechanistic link.

**Part III – Minor Issues: Editorial and Data Presentation Modifications**

Reviewer #1: None

Reviewer #2: I have no further comments

Reviewer #3: 1. The discussion would benefit from slightly tightening the contrasting roles of IFI27 on IFN signaling vs. PKR activation—especially to help readers reconcile the previous immunosuppressive roles with its proposed PKR activation.

2. In the methods section, the RNAbindRPlus prediction of RNA-binding sites in IFI27 could be briefly described with confidence scores or thresholds.

3. Consider standardizing the naming of IFI27 mutants (e.g., IFI27-S63L vs. S63L variant) for clarity.

4. the IFI27-PACT interaction remains inferred as RNA-dependent but would benefit from RNase treatment during PKR co-IPs, not just PACT co-IPs. Nonetheless, the authors do not claim a direct interaction and have added poly(I:C)-dependent modulation data, which strengthens their model.

PLOS authors have the option to publish the peer review history of their article (what does this mean? ). If published, this will include your full peer review and any attached files.

**Do you want your identity to be public for this peer review?** For information about this choice, including consent withdrawal, please see our Privacy Policy .

Reviewer #1: No

Reviewer #2: No

Reviewer #3: No

**Figure resubmission:**
---

## [Editor Report · Decision Letter 2]

Dear DR DeDiego,

We are pleased to inform you that your manuscript 'IFN alpha inducible protein 27 (IFI27) acts as a positive regulator of PACT-dependent PKR activation after RNA virus infections' has been provisionally accepted for publication in PLOS Pathogens.

Best regards,

Peter Sarnow

Academic Editor

PLOS Pathogens

Alexander Gorbalenya

Section Editor

PLOS Pathogens

Sumita Bhaduri-McIntosh

Editor-in-Chief

PLOS Pathogens

orcid.org/0000-0003-2946-9497

Michael Malim

Editor-in-Chief

PLOS Pathogens

orcid.org/0000-0002-7699-2064
---

## [Editor Report · Acceptance letter]

Dear DR DeDiego,

We are delighted to inform you that your manuscript, "IFN alpha inducible protein 27 (IFI27) acts as a positive regulator of PACT-dependent PKR activation after RNA virus infections," has been formally accepted for publication in PLOS Pathogens.

Best regards,

Sumita Bhaduri-McIntosh

Editor-in-Chief

PLOS Pathogens

orcid.org/0000-0003-2946-9497

Michael Malim

Editor-in-Chief

PLOS Pathogens

orcid.org/0000-0002-7699-2064